# Characterisation of between-cluster heterogeneity in malaria cluster randomised trials to inform future sample size calculations

Joseph Biggs [1] ✉, Joseph D. Challenger [2], Dominic Dee [2], Eldo Elobolobo [3], Carlos Chaccour [4,5,6], Francisco Saute[3], Sarah G. Staedke[7,8], Sibo Vilakati[9], Jade Benjamin Chung [10,11], Michelle S. Hsiang[11,12,13], Edgard Diniba Dabira[14], Annette Erhart[14], Umberto D'Alessandro[14], Rupam Tripura[15,16], Thomas J. Peto [15,16], Lorenz Von Seidlein [15,16], Mavuto Mukaka[15,16], Jacklin Mosha[17], Natacha Protopopoff [18,19], Manfred Accrombessi[20,21], Richard Hayes[1], Thomas S. Churcher [2] & Jackie Cook[1]

Cluster randomised trials (CRTs) are important tools for evaluating the community-wide effect of malaria interventions. During the design stage, CRT sample sizes need to be inflated to account for the cluster heterogeneity in measured outcomes. The coefficient of variation (k), a measure of such heterogeneity, is typically used in malaria CRTs yet is often predicted without prior data. Underestimation of k decreases study power, thus increases the probability of generating null results. In this meta-analysis of cluster-summary data from 24 malaria CRTs, we calculate true prevalence and incidence k values using methods-of-moments and regression modelling approaches. Using random effects regression modelling, we investigate the impact of empirical k values on original trial power and explore factors associated with elevated k. Results show empirical estimates of k often exceed those used in sample size calculations, which reduces study power and effect size precision. Elevated k values are associated with incidence outcomes (compared to prevalence), lower endemicity settings, and uneven intervention coverage across clusters. Study findings can enhance the robustness of future malaria CRT sample size calculations by providing informed k estimates based on expected prevalence or incidence, in the absence of cluster-level data.

To inform malaria control and elimination policy, the World Health Organisation (WHO) relies on cluster randomised trials (CRTs) to evaluate the effectiveness of interventions in the community[1]. Malaria control tools, including insecticide-treated nets (ITNs), vaccines, insecticidal spraying and chemoprevention are typically implemented at the community level, and can benefit individuals directly and indirectly[2–7]. CRTs assess the effectiveness of such tools by randomising groups (clusters) into intervention and control arms, and estimating effect size(s) by comparing them[8]. However, individual malaria outcomes within clusters (such as households, schools and villages), are often highly correlated due to shared exposure to similar risk factors, more so than between clusters[9,10]. Such correlation increases

the variability between clusters, requiring larger sample sizes to maintain statistical power (the probability of detecting a statistically significant difference if a true difference exists). Cluster heterogeneity thereby exacerbates logistical challenges and costs associated with CRTs[8].

Sample size calculations for CRTs should account for the heterogeneity caused by correlations between clusters. This requires incorporating estimates of either the coefficient of variation (k) or the intracluster correlation coefficient (ICC). K represents an absolute measure of between-cluster variability, calculated as the ratio of the standard deviation of cluster-level outcomes to the overall mean outcome. In contrast, the ICC is a relative measure, quantifying the proportion of total variance in trials attributable to between-cluster variation[11,12]. In the absence of site-specific data, trialists often approximate these values during the design phase. If underestimated, trials risk being underpowered; if overestimated, they may become overpowered, leading to a waste of resources. Inaccurate classification of cluster heterogeneity is compounded by frequent omission of empirical estimates of k or ICC in trial publications[13–15], despite being a requirement in CONSORT guidelines[16]. Prior to this study, our systematic review of malaria CRTs showed that 80% of trials used k to account for cluster heterogeneity in their sample size calculations, while only 20% provided retrospectively calculated empirical estimates according to trial data. Among the trials that did, large disparities were observed between predicted and empirical values[17].

Malaria, a vector borne, parasitic disease transmitted by female Anopheles mosquitoes, causes significant morbidity and mortality globally[18]. Transmission is influenced by environmental and human behavioural factors, leading to spatio-temporal variation in risk across geographical areas[19,20]. During wet seasons, increased rainfall creates more mosquito breeding sites, which amplify vector populations and intensify transmission risk[21–23]. In addition, as risk in the community decreases, malaria transmission becomes geographically more focal due to increased heterogeneity in vector breeding sites, immunity, human behaviours and malaria intervention effectiveness[24–27]. Such heterogeneity in malaria transmission across geographical regions likely translates to heterogeneity in malaria outcomes between study clusters.

Previous studies have investigated cluster heterogeneity patterns associated with different malaria outcome metrics. In Southeast Asia, a secondary analysis of a multi-country malaria CRT highlighted how empirical ICC estimates were influenced by country, Plasmodium species and type of outcome measure (prevalence or incidence), although the authors speculate that this variation could be due in part to chance given the low cluster numbers[10]. In Namibia, a secondary analysis of a malaria CRT showed that sensitive serological endpoints measuring previous exposure to malaria generated comparable effect size estimates to outcomes based on PCR (polymerase chain reaction assay) endpoints from the same individuals, but exhibited lower between-cluster heterogeneity. The authors suggest that this may be due to serological testing capturing both current and recently exposed cases, which are likely more homogeneously distributed across geographical regions than current cases detected solely by PCR[28].

Studies have also explored cluster heterogeneity patterns for given malaria outcomes. In the Gambia, a malaria CRT showed empirical k estimates varied significantly between study arms and years, often exceeding the predicted value[29]. In Tanzania, a CRT secondary analysis highlighted the heterogeneity in prevalence ICC estimates between repeated surveys, which authors speculate reflect seasonal fluctuations in malaria and waning effects of interventions[30]. Lastly in Nigeria, a study showed that reductions in malaria prevalence from 2010 to 2015 were associated with increased between-state variability, highlighting the relationship between transmission intensity and focality[31].

Previous findings underscore the need to better characterise cluster heterogeneity in malaria CRTs and understand factors that are associated with it. To address this, we conducted a meta-analysis of cluster-level data from previous malaria CRTs measuring epidemiological outcomes (prevalence or incidence) to: (1) estimate empirical values of k, (2) assess the impact of cluster heterogeneity on study power and effect size uncertainty, and (3) identify factors associated with cluster heterogeneity. These insights are expected to improve future CRT design, ensuring robust evaluation of malaria interventions.

## Results
Of the 71 malaria cluster-randomised trials (CRTs) identified in our previous systematic review, we obtained cluster-level epidemiological data from 24 trials (Supplementary Table 1). These parallel CRTs, conducted across 21 different countries between 2000 and 2021, evaluated various malaria interventions, including vector control (67%, 16/24 trials) and chemoprevention (25%, 6/24 trials). Most trials featured two study arms (71%, 17/24 trials; range: 2–4 arms). Cluster-level prevalence and incidence data were provided by 19 and 14 of the 24 trials, respectively (Supplementary Table 2). The characteristics of trials in this meta-analysis closely resembled those from the previous systematic review, suggesting they form a representative sample (Supplementary Table 3).

Characteristics of the prevalence data provided by trials are shown in Supplementary Table 4. In total, cluster-level prevalence data were available from 57 cross-sectional surveys (range per trial: 1–7) spanning 816 clusters (range per trial: 6–104 clusters). The average number of individuals surveyed per cluster ranged from 8.7 to 1,064. Prevalence outcomes were measured using PCR (Polymerase chain reaction assays), RDTs (rapid diagnostic tests), or microscopy. Cluster-level intervention coverage data were provided for 8/19 trials with prevalence data. Among trials that provided prevalence data, 13/19 trials determined the numbers surveyed according to sample size calculations that accounted for cluster heterogeneity. According to control arm prevalence throughout the trials, 5/19 trials were categorised as high endemicity, 8/19 were classified as medium and 6/19 were categorised as low.

Characteristics of the incidence data obtained from 14 trials are shown in Supplementary Table 5. Eight trials provided incidence data generated from active case detection (ACD), 5 from passive case detection (PCD) and one trial collected separate incidence measures using ACD and PCD. Cluster-level incidence data were collected from a total of 751 clusters (trial range: 6-187) and were totalled for each study year. Most trials (11/14) provided a sample size justification for the number of individuals enroled to estimate a difference in incidence between arms. Based on control-arm incidence throughout each trial, 4/14 trials were categorised as high endemicity, 3/14 were considered medium and 7/14 were classified as low.

### Characterisation of outcome between-cluster heterogeneity in malaria CRTs
We characterised the between-cluster heterogeneity of outcomes at the survey-arm level for prevalence outcomes and at the study-year arm level for incidence outcomes (Fig. 1a). The overall survey-arm prevalence for all trials ranged between 0% and 82.7% (median: 25.6%) while the overall annual malaria incidence per person for each study year-arm ranged from <0.01 to 7.19 malaria cases per person year (py) (median: 0.22/py) (Fig. 1b). At the cluster level, prevalence ranged between 0% and 100% (median: 25%) and incidence ranged from 0 to 15.2/py (median: 0.24 py) (Fig. 1c). The cluster-level distribution of prevalence and incidence outcomes for each trial is shown in Supplementary Fig. 1. There was good agreement observed between the methods-of-moments and regression approach for estimating prevalence and incidence k (Supplementary Fig. 2). As the regression

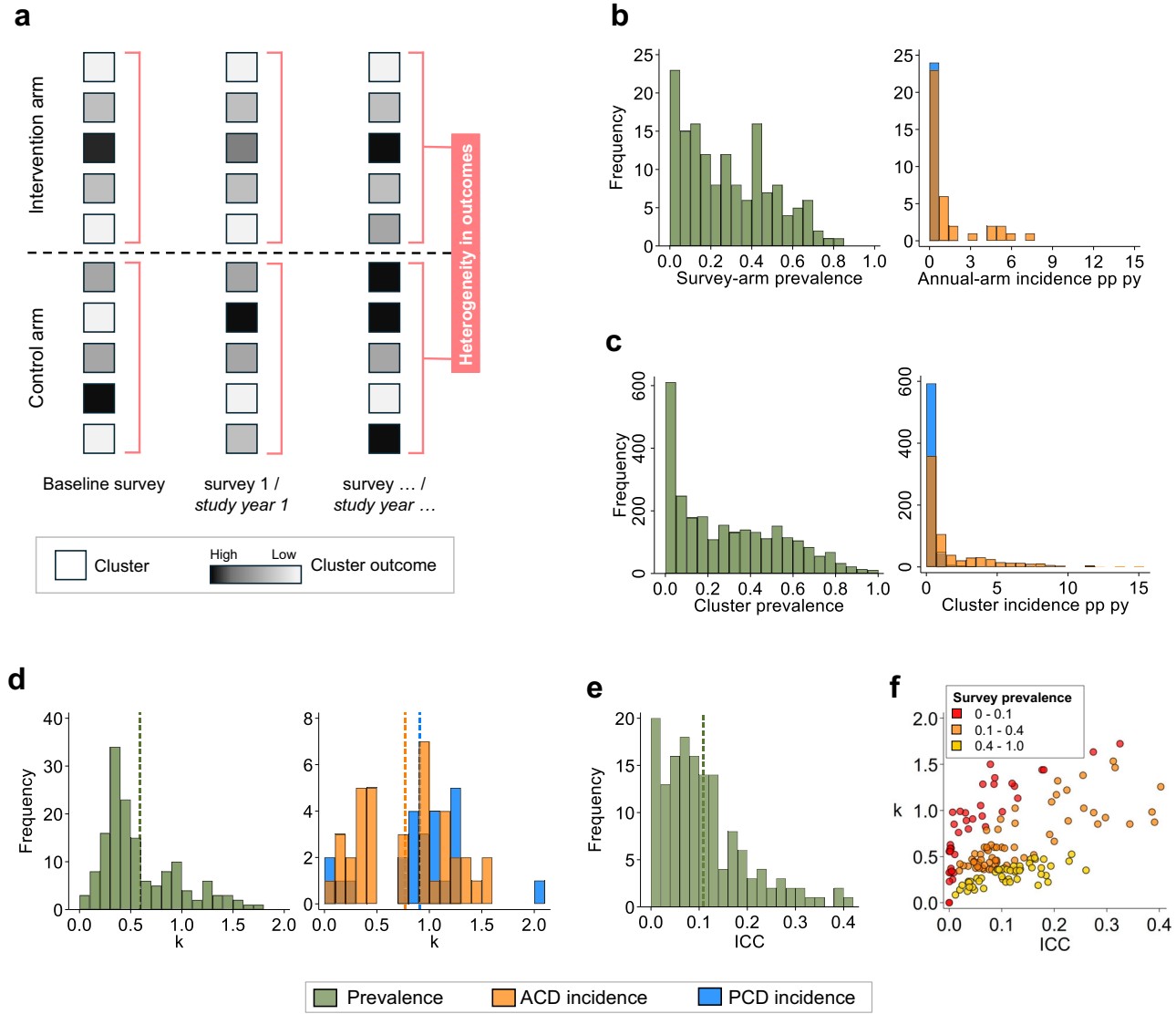

**Fig. 1 | Characterisation of between-cluster heterogeneity of outcomes in malaria CRTs. a** Schematic representation of between-cluster heterogeneity in prevalence at the survey-arm level and incidence at the study-year arm level. **b** Distribution of the overall survey-arm prevalence and study-year incidence among all trials. **c** Distribution of the cluster-level prevalence and incidence among all trials. **d** Distribution of empirical prevalence k estimates for each survey-arm and incidence k estimates for each study year-arm among trials. Vertical dashed line: median k estimate. **e** Distribution of empirical prevalence ICC estimates for each survey-arm among trials. Vertical dashed line: median ICC estimate. **f** Scatter plot comparing prevalence k and ICC estimates at the survey-arm level stratified by the overall prevalence of the corresponding surveys. ACD: active case detection incidence. PCD: passive case detection incidence. pp py: per person per year.

approach can be used to estimate k 95%CIs, this method was used in all subsequent analyses. Among all survey-arms in all trials, prevalence k ranged from <0.01 to 1.72 (median: 0.46), while in all study year-arms in all trials, incidence k ranged between <0.01 and 2.05 (median: 0.91). Overall among all trials, PCD incidence k (median: 0.97) was higher than ACD incidence k (median: 0.84) (Fig. 1d). In addition to k, we estimated the ICC for cluster-level prevalence at the survey-arm level (Fig. 1e). Prevalence ICC ranged between <0.01 and 0.40 with a median of 0.09. As k and ICC represent distinct measures of between-cluster variability, we compared them at the survey-arm level (Fig. 1f). Among survey-arms with a prevalence >10%, we observed a positive correlation between k and ICC. In contrast, among survey-arms with overall prevalences <10%, larger disparities were observed between k and ICC. When ICC estimates were near zero, k often exceeded 0.7.

We next investigated whether between-cluster heterogeneity differed between study arms of trials (arm-differential between-cluster heterogeneity). For each survey and study year of each trial, we estimated the difference in k and ICC between arms and compared differences against the trial period. For prevalence, k values were typically higher in the intervention arm during the post-intervention period (Fig. 2a) while ICC estimates were often larger in the control arm (Fig. 2b). This pattern was impacted by the overall survey prevalence which showed the difference in k between arms was lower in high prevalence surveys while the ICC difference was lower in low prevalence surveys (Supplementary Fig. 3). For incidence outcomes, k was typically higher in control arms of trials (Fig. 2c). Despite no observed clear arm-differential between-cluster heterogeneity patterns among trials, k and ICC estimates were rarely similar between arms. In addition to arm-differential patterns, we also explored temporal patterns in between-cluster heterogeneity during trials. For prevalence outcomes, k estimates in the control arms of repeated surveys among trials were lower and temporally more stable in high endemicity trials compared medium and low endemicity trials (Fig. 2d). This pattern was similar for incidence outcomes and intervention clusters among prevalence surveys (Supplementary Fig. 4a–c). In contrast, prevalence ICC estimates

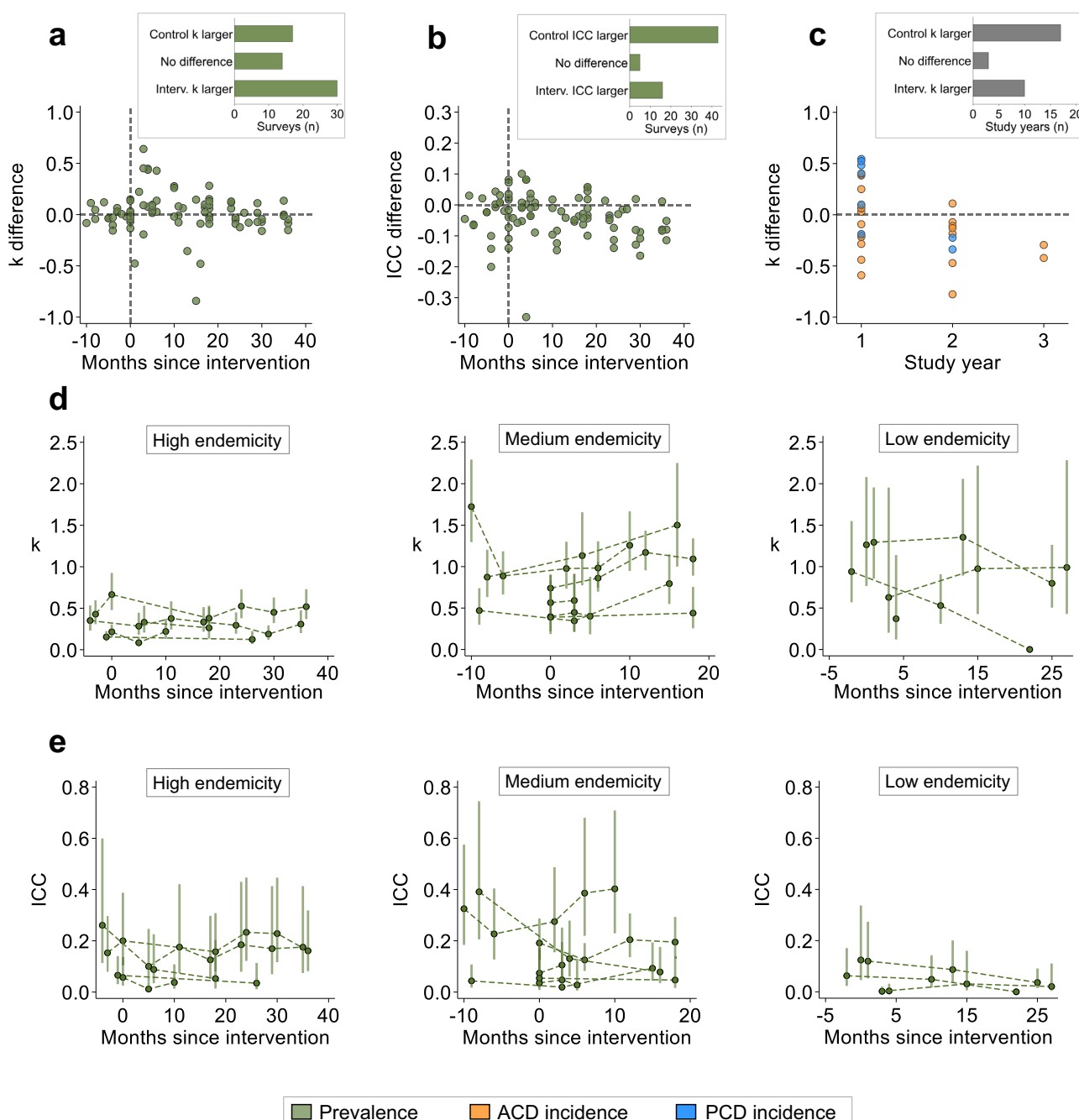

**Fig. 2 | Arm-differential and temporal patterns in between cluster hetero-geneity in malaria CRTs. a**, **b** Difference in prevalence k (**a**) and ICC (**b**) by arm for each survey by months since intervention among trials. Positive k/ICC difference (above horizontal dashed line) indicates higher k/ICC value in the intervention arm compared to the control. Horizontal dashed line: Time of intervention imple-mentation. Bar chart: number of surveys with k/ICC estimates higher in the inter-vention arm (>10% difference), the same between arms (<10% difference) and higher in the control arm (>10% difference). **c** Difference in incidence k between arms for each by study year among trials. Positive k difference (above horizontal

dashed line) indicates higher k value in the intervention arm compared to the control. Bar chart: number of study years with k estimates higher in the interven-tion arm (>10% difference), the same (<10% difference) and higher in the control arm (>10% difference). **d**, **e** Temporal patterns in control-arm prevalence k (**d**) and ICC (**e**) values for each survey over months since intervention implementation, stratified by trial endemicity. Dots represent control-arm survey k estimates. Dashed lines represent individual trials. Error bars represent 95%CIs. Surveys conducted <0 months since intervention represent those conducted prior to intervention roll out.

among the control arms of repeated surveys were largest and more temporally variable in the trials conducted in medium endemicity settings (Fig. 2e). A trend similarly observed for intervention-arm ICC estimates (Supplementary Fig. 4d). Together, results illustrate that cluster-outcome heterogeneity changes over the course of malaria CRTs, regardless of intervention presence, but tends to vary less in trials conducted in high endemicity settings.

## Impact of empirical between-cluster heterogeneity on trial power and effect size precision

Among trials that were powered to detect a difference in incidence and/or prevalence between arms (prevalence: 13, incidence: 11, Sup-plementary Tables 4 and 5), all used k in their sample size calculations to account for clustering effects. We therefore compared observed prevalence k values for each survey-arm, and incidence k values for

each study year, to k values predicted in each trials original sample size calculation (Fig. 3a). Assuming trialists anticipated their predicted k estimates would remain constant throughout their trials, 72.5% (29/40) of prevalence k and 57.9% (11/19) of incidence k values were underestimated. For each prevalence survey and incidence year, we compared the observed k in the control arms to observed power (%) based on empirical k and control-arm prevalence or incidence (Fig. 3b). Prevalence surveys or incidence years with elevated k values had reduced power to detect their predicted effect size(s). To determine whether trials were adequately powered at the beginning of each trial (>80%), we recalculated power according to predicted and observed parameters: baseline control-arm prevalence/first year control incidence and control arm k values. Results showed that 50% (6/12) of trials that measured prevalence, and 55% (6/11) of trials that measured incidence, achieved <80% power at the start (Fig. 3c).

In addition to power, we investigated the impact of empirical between-cluster heterogeneity on effect size precision. Among all post-intervention surveys and study years, we compared empirical control-arm k estimates to observed arm-level effect sizes and corresponding 95%CIs. Elevated between-cluster heterogeneity was associated with decreased precision around prevalence and rate ratios (Fig. 3d). Similarly to arm-level effect sizes, we explored the impact of empirical k estimates on cluster-level effect sizes (intervention cluster outcome / overall outcome in the corresponding control arm). For prevalence outcomes, among surveys with lower k estimates (<0.3), intervention cluster-effect sizes were normally distributed below a prevalence ratio of 1. In contrast, among surveys with higher k estimates (>1.2), cluster-level effect sizes exhibited a zero-inflated right-skewed distribution, indicating intervention clusters exhibited either a very large or no difference compared to the mean control arm prevalence (Fig. 3e). A similar pattern was observed for incidence outcomes (Fig. 3f). These results demonstrate large between-cluster heterogeneity in outcomes is equivalent to large between-cluster variability in treatment effects.

We examined the magnitude of effect sizes and size of trials required to accommodate such elevated k estimates observed in malaria CRTs. For a hypothetical trial with 20 clusters per arm, a cluster size of 50, a k estimate of 1.2, and a control prevalence of 10%, such a trial would only be adequately powered (80%) to detect a minimum effect size of 0.8 (i.e a prevalence of 2% in the intervention arm) (Fig. 4a). To detect smaller effect sizes (<40%) between arms, very large numbers of clusters (>150 per arm) would be required at 80% power with k values >1 (Fig. 4b).

### Factors associated with between-cluster heterogeneity
Given the detrimental impact of large between-cluster heterogeneity in malaria CRTs, we explored factors that influence k. Firstly, we investigated whether larger k values were more common with prevalence or incidence outcomes. Among malaria CRTs that measured both incidence and prevalence during overlapping time periods ($n = 9$) (Supplementary Table 2), the data show that control-arm prevalence k estimates were lower than incidence k values for 89% (8/9) of trials (Supplementary Table 6). In addition to type of outcome measure, we explored whether other trial covariates were associated with elevated k values ($k > 0.5$) using random effects logistic regression. For prevalence outcomes, decreasing survey prevalence and surveys conducted in the malaria season were associated larger k estimates ($p < 0.05$) (Supplementary Table 7). Due to lower number of study years in this meta-analysis, we were unable to replicate this analysis for incidence outcomes.

To further characterise the relationship between k and overall survey prevalence or study year incidence, we fitted a linear regression model using log-transformed estimates of k to account for the non-linear association (Fig. 5a, b). As most study-years had very low overall passive incidence estimates, we did not include the PCD incidence data. For prevalence outcomes, increasing survey-arm prevalence was associated with decreasing k and k uncertainty (Fig. 5a). According to our model, survey-arms with an overall prevalence of 20% had a predicted k value of 0.60 [95%CI: 0.55–0.65] while an overall prevalence of 60% had a predicted k estimate of 0.26 [95%CI: 0.23–0.29]. Likewise with active incidence outcomes, among study years with overall incidences of 0.2/py and 1.2/py, predicted k values were 0.94 [95%CI: 0.45–1.44] and 0.19 [95%CI: 0.02–0.36], respectively (Fig. 5b).

We stratified the relationship between k and survey prevalence by survey season (malaria vs non-malaria). Results showed malaria season surveys were associated with higher k values than non-malaria season surveys in low-prevalence settings (<30%) (Fig. 5c–e). We also explored the impact of survey seasonality on effect size. Among the 33% (8/24) of trials that conducted cross-sectional surveys in both malaria and non-malaria seasons, 38% (3/8) experienced larger effect sizes, and had higher k values, in malaria season surveys compared to non-malaria-season surveys (Supplementary Fig. 5). Regarding coverage of malaria interventions, we found that overall survey intervention coverage had no apparent effect on prevalence k (Fig. 5f). Nonetheless, increasing between-cluster heterogeneity in intervention coverage (intervention coverage k) was associated with increased prevalence k (Fig. 5g). Moreover, increasing intervention coverage k was associated with larger degrees of uncertainty around observed effect size estimates (Fig. 5f). This suggests uneven intervention coverage across clusters is associated with between-cluster variability in prevalence, which can result in decreased effect size precision.

## Discussion
Results from this meta-analysis of 24 malaria CRTs highlight that between-cluster heterogeneity of epidemiological outcomes is often large, different between study arms and temporally variable. When study power was recalculated using empirically derived estimates of the coefficient of variation (k), many trials were found to have had a low probability of detecting statistically significant differences between arms. Moreover, large k values were associated with reduced effect size precision. Here, we identified factors that influence k in malaria CRTs, which could be used to reduce large heterogeneity in future trials, including choice of outcome measure, endemicity of the chosen site, seasonality in transmission and uneven intervention coverage across clusters. By carefully considering these factors, future malaria CRT design can be optimised to help ensure trials are adequately powered and more statistically robust. A summary of our recommendations and considerations based on the study findings are presented in Box 1.

In our previous systematic review of 71 epidemiological malaria CRTs, we highlighted how approximately 70% of trials used predicted coefficient of variation values in their power/sample size calculations[17]. Here, we show that most predicted k values were underestimated and that empirical values often exceeded conservative estimates deemed appropriate for infectious disease CRTs[32]. As prior or baseline cluster-level data, which are needed to obtain empirical estimates of k, are frequently unavailable before sample size estimation[8,11], we provide suitable k values for given endemicity settings. If trialists obtain reliable predictions of overall incidence or prevalence in their trial setting, our model can be used to provide model-informed k values that can be incorporated into sample size/power calculations. In high endemicity settings we observed low values of k which mirror estimates for other infectious disease CRTs[8,33,34]. Contrastingly, in low endemicity settings, large and temporally unstable k estimates were identified which are problematic for trial design. If used in trial sample size calculations, large effects sizes and/or excessive numbers of clusters would be required to achieve adequate power. Major logistical and financial constraints associated with large CRTs means that increasing trial size likely represents an unsustainable solution to large between-cluster variability[1,8,35]. However, achieving larger relative reductions in

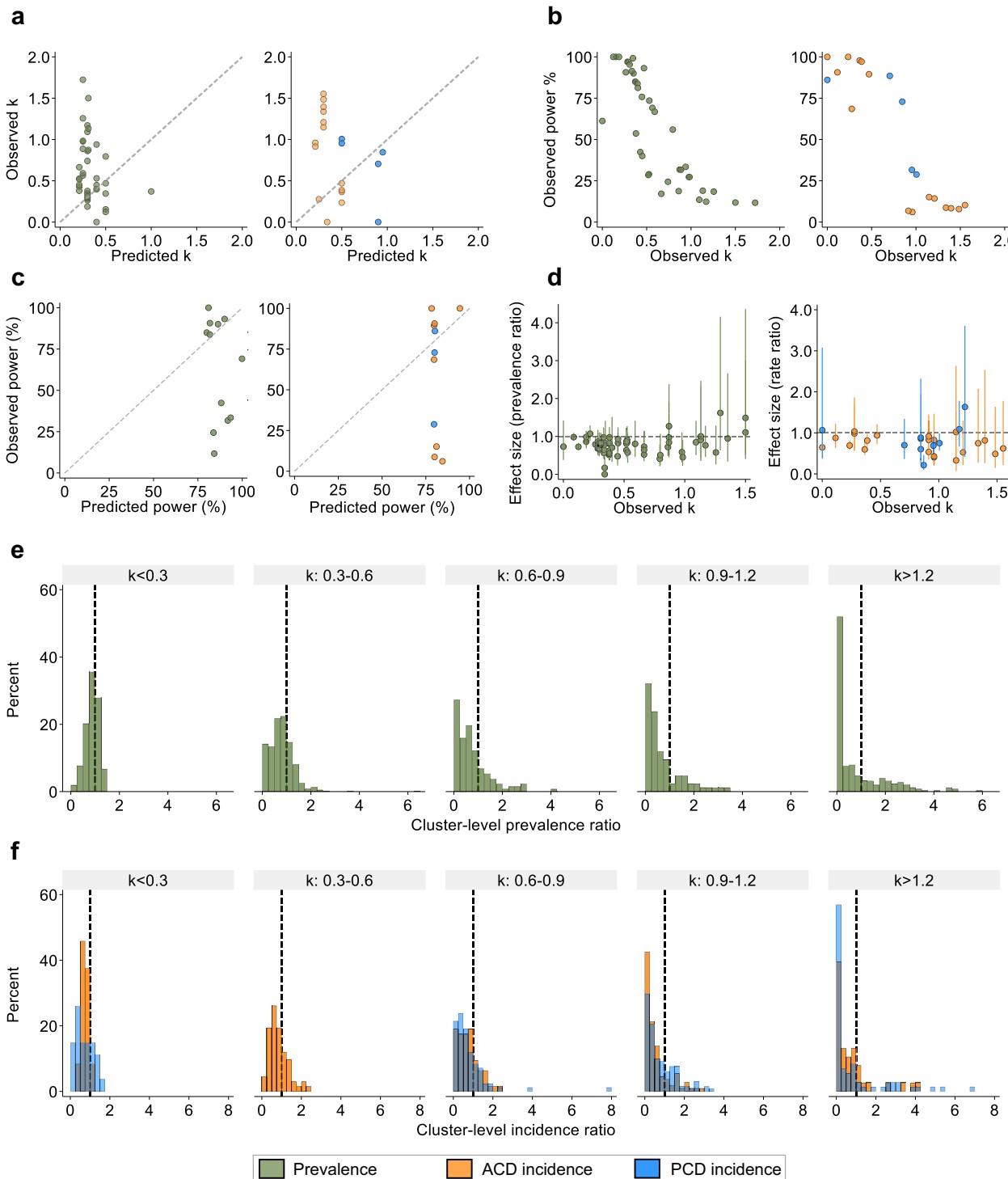

**Fig. 3 | The impact of observed between-cluster heterogeneity on study power and effect size precision. a** Scatter plot comparing the empirical prevalence k estimates for each control-survey arm (green) and incidence k estimates for each control-arm study year (orange: ACD, blue: PCD incidence) to k estimates predicted in the trials original sample size/power calculations. Dashed line represents equality. **b** Scatter plot comparing the recalculated study power for each post-intervention survey (green) or study year (orange/blue) to detect the predicted effect size according to empirical k estimates and control arm prevalence or incidence values against corresponding empirical k estimates. All remaining sample size parameters remained identical to original calculations. **c** Scatter plot comparing the original power of trials against the observed study power according to empirical control-arm estimates of k and prevalence (green) or incidence (orange: ACD, blue: PCD) in the baseline surveys or first years of trials, respectively. Dashed line represents equality. **d** Scatter plot comparing the empirical control-arm

prevalence k (green) or incidence k (orange: ACD, blue: PCD) for each post-intervention survey or study year against the corresponding arm-level effect size estimate (dots) and 95%CI (error bars) between arms. Horizontal dashed line represents effect size of 1. **e** Histogram of all the post-intervention surveys cluster-level effect sizes stratified by control survey k estimates. Cluster-level effect size: intervention cluster prevalence/corresponding control arm average prevalence. Horizontal dashed line represents the null effect size of one. Intervention clusters with prevalence ratios <1 exhibited a prevalence value less than the average prevalence in the control. **f** Histogram of all study years cluster-level effect sizes stratified by the control year k estimates. Cluster-level effect size: intervention cluster incidence/corresponding control arm average incidence. Horizontal dashed line represents the null effect size of one. Intervention clusters with incidence ratios <1 exhibited an incidence value lower than the average incidence in the control.

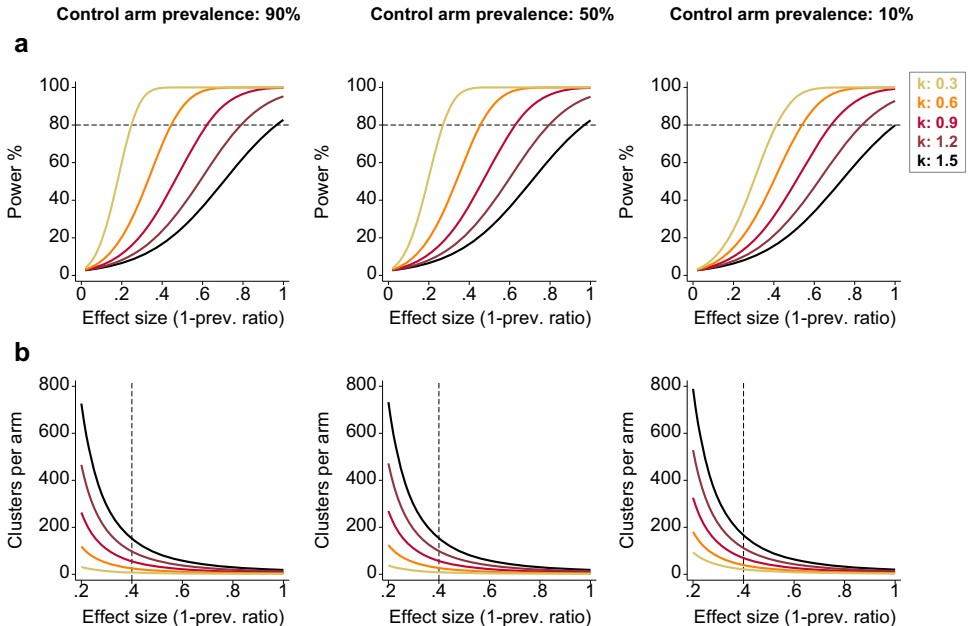

**Fig. 4 | Impact of varying between-cluster heterogeneity and effect sizes on study power and trial size. a** Estimated study power according to varying k estimates (range: 0.3–1.5), effect sizes (1-prevalence ratio, range: 0–1) and control arm prevalences (90%, 50% and 10%) for a hypothetical CRT. **b** Estimated sample size (required clusters per arm) according to varying k estimates (range: 0.3–1.5), effect sizes (1-prevalence ratio, range: 0.2–1) and control arm prevalences (90%, 50% and 10%) for a given number of individuals surveyed per cluster in a hypothetical CRT.

outcomes between arms may be more feasible in trials with lower prevalence/incidence, and large effect sizes are often anticipated in such settings[17]. Given novel malaria elimination strategies still require community evaluation, trialists must balance realistic effect size expectation and public health relevance against potentially large cluster heterogeneity when designing trials in low-endemicity settings. Moreover, as malaria transmission can be spatially and temporally variable, particularly in low endemicity settings[19–21,25], accurately predicting study-area prevalence/incidence to derive informed estimates of k remains challenging.

Results also showed that between-cluster heterogeneity was rarely similar between study arms. We propose two main explanations for this observation. First, k estimates are sensitive to small changes in overall prevalence or incidence, particularly in low transmission settings where the denominator approaches zero[36]. Second, the malaria CRTs included in this review evaluated a range of intervention types, which may have induced either homogeneous or heterogeneous treatment effects across clusters[37]. Notably, for prevalence outcomes in intervention arm, the ICC was generally lower, while k was often higher compared to the control arm. This may reflect more consistent intervention use within clusters, promoting within-cluster homogeneity and lowering the ICC, while the reduction in mean prevalence in the intervention arm may have inflated k estimates[36]. Regardless, commonly used analytical methods in CRTs, including random effects regression modelling and generalised estimating equations, typically presume equal cluster variability across treatment arms[32]. Whether malaria CRTs should consider analysis methods that allow for arm-specific variances remains an area of continued investigation, however recent evidence suggests that standard methods remain robust despite differences in ICC values between arms[38].

To further overcome large-between cluster heterogeneity in outcomes in malaria CRTs, trialists could consider modifying other controllable factors. Overall, incidence outcomes showed a higher degree of between-cluster variability than prevalence outcomes among the same trials measured over similar time periods, as was also shown previously in southeast Asia[10]. We suggest this is in part due to the characteristics of these different

outcomes. Cluster-level incidence, unlike prevalence, is only bound by zero which allows for more skewed cluster distributions that might exacerbate k estimates[39]. Moreover, incidence measures are highly variable due to contrasting definitions of new cases and follow-up time adjustments among trials[40], potentially making the aggregated endpoint highly variable across clusters. Prevalence outcomes therefore may be an easier endpoint measure to power for in malaria CRTs.

Surveys conducted in, or shortly after, rainy seasons were associated with elevated between-cluster heterogeneity compared with dry season surveys. This aligns with the idea that regional differences in geography and human behaviour, coupled with increased rainfall, contribute to spatially uneven amplification of malaria transmission intensity[21,41]. It should be noted that rainy season surveys in some malaria CRTs were associated with larger effect sizes which likely compensated for the loss of power due to elevated between-cluster variability. Lastly, we highlight that a potential driving force for high between-cluster heterogeneity in prevalence was uneven intervention coverage across clusters during the implementation period. Triallists often strive for maximum intervention coverage to achieve their predicted effect sizes[8,32], although we suggest that for a given overall coverage, aiming for uniform coverage across clusters may assist in maintaining power.

Findings from this meta-analysis emphasise that further research is necessary to ensure future malaria tools are effectively and sustainably evaluated in the community. As we strive towards malaria elimination, more cutting-edge malaria interventions will require trial evaluation in low endemicity settings which could be hindered by large-between cluster heterogeneity. Alternative and adaptive CRT designs, including cluster stratification, matching and sample size re-estimation[8], may help to minimise between-cluster heterogeneity and maintain power. However this will need investigating across different endemicity settings due to varying spatial and temporal heterogeneity in transmission[12]. Using more sensitive diagnostics may prove a suitable strategy to minimise k. In our meta-analysis, most trials used one type of diagnostic to capture malaria cases. Consequently, we were unable to determine whether

accurately capturing low-density malaria infections reduced cluster heterogeneity as demonstrated in Namibia[28].

There are limitations associated with our research findings. Firstly, estimates of prevalence and incidence are not always comparable between trials, due to the use of different diagnostics and/or age ranges tested. Secondly, survey seasonality was crudely categorised according to information given in journal publications and may not have been reflective of the intensity of rainfall in trial settings. Thirdly, as only a minority of trials provided intervention coverage data, the association between prevalence and intervention coverage between-cluster heterogeneity is driven by a small number of trials. Lastly, effect size estimates generated in this analysis were restricted to cluster-level analyses as we only obtained cluster-level data. Consequently, effect size estimates likely differ slightly from original trial analyses that often utilised individual-level analyses, however, it is well documented that increased between-cluster variability reduces precision of both cluster-level and individual-level effect size estimates[32] so the findings would likely still be similar.

Large between-cluster heterogeneity of epidemiological outcomes observed in malaria CRTs represents a major challenge for the evaluation of community-wide interventions. If future trials fail to overcome the impacts of between-cluster heterogeneity, the effects of vital interventions against malaria could be missed. Future research is needed to identify design and analysis strategies that can ensure trials can effectively and sustainably evaluate novel interventions which are key to eliminate malaria globally.

## Methods

### Trial data

We sought cluster-level malaria outcome data from corresponding authors of published CRTs identified in our previous systematic review (PROSPERO: CRD42022315741)[17]. Authors were initially approached by email, with a second follow-up email to non-responders. The initial review included 71 malaria CRTs that qualified for inclusion if they measured malaria-specific, epidemiological outcomes (prevalence or incidence) and randomised at least six geographical clusters to study arms. For trials measuring prevalence, we requested the number of malaria-positive individuals and total tested per cluster, study arm, and survey (Supplementary Table 8). For incidence measured via active case detection (ACD), we requested new malaria cases and total person-years at risk, and for passive case detection (PCD), new malaria cases and the population at risk, stratified by cluster, study arm, and trial year (Supplementary Table 9, 10). These data were supplemented with covariates from published articles, including diagnostic method and age range tested. Malaria prevalence in this study referred to the number of individuals tested positive for malaria over the total number of individuals tested for malaria. Malaria incidence in this study referred to the number of new cases divided by the total person years at risk (ACD) or total population in each cluster at risk (PCD). We classified trial endemicity based on control-arm prevalence or incidence averaged during the entire trial. Trial endemicity according to prevalence was categorised as high (>40%), medium (10–40%), or low (0–10%). For incidence, trial endemicity was categorised by malaria cases per person-year (py) as high (>0.8/py), medium (0.2–0.8/py), or low (0–0.2/py).

Trial prevalence data were further supplemented with requested intervention coverage/usage data (number of intervention users or individuals covered by interventions/total number surveyed) stratified by cluster, arm and survey. According to the month interventions were deployed and survey dates, we calculated the months since intervention(s) were introduced for each survey and categorised surveys as pre/post-intervention. We categorised all trial surveys as malaria season surveys if they were conducted within the publication-stated rainy season, plus one month to account for the delay between rains and vector propagation[22,41]. Surveys administered outside this range were considered non-malaria season surveys.

### Between-cluster heterogeneity estimation

Methods-of-moments and mixed effects regression modelling approaches were used to estimate empirical values of prevalence k and ICC at the survey-arm level and incidence k at the study year-arm level according to methods described by Hayes and Moulton[12,32]. We refrained from estimating incidence ICC values, as rates with person-time denominators lack a clearly defined unit of observation[36].

For the methods-of-moments approach, we computed the empirical variance ($s^2$) of each survey arm for cluster-level prevalence (Eq. 1) and each study year-arm for cluster-level incidence (Eq. 2) according to:

$$Prevalence \ \ s^2 = \frac{\sum (p_i - \bar{p})^2}{c - 1} \tag{1}$$

$$Incidence \ \ s^2 = \frac{\sum (r_i - \bar{r})^2}{c - 1} \tag{2}$$

where $c$ refers to the total number of clusters, $p_i$ is the malaria prevalence in the $i$th cluster and $\bar{p}$ represents the mean cluster prevalence ($\sum p_i / c$). For incidence, $r_i$ represents the annual malaria incidence per person in the $i$th cluster and $\bar{r}$ represents mean incidence across clusters ($\sum r_i / c$).

To estimate the true between-cluster variance $\hat{\sigma}_B^2$ for each survey arm for prevalence (Eq. 3) or study year-arm for incidence (Eq. 4), we subtracted the random sampling error from the empirical variance $s^2$ as follows:

$$Prevalence \ \hat{\sigma}_B^2 = s_{prev}^2 - \frac{p(1-p)}{\bar{n}_H} \tag{3}$$

$$Incidence \ \hat{\sigma}_B^2 = s_{inci}^2 - \frac{r}{\bar{f}_H} \tag{4}$$

where $p$ refers to the overall survey-arm malaria prevalence, $\bar{n}_H$ is the harmonic mean of the total number of individuals $n_i$ tested per cluster ($c / \sum \left(\frac{1}{n_i}\right)$), $r$ refers to the overall study year-arm malaria incidence and $\bar{f}_H$ is the harmonic mean of the total follow up time in years $y_i$ per cluster ($c / \sum \left(\frac{1}{y_i}\right)$).

We then estimated prevalence k for each survey-arm (Eq. 5) and incidence k for each study year arm (Eq. 6) according to:

$$Prevalence \ \hat{k} = \frac{\hat{\sigma}_{B \ prev}}{p} \tag{5}$$

$$Incidence \ \hat{k} = \frac{\hat{\sigma}_{B \ inci}}{r} \tag{6}$$

In addition to the methods-of-moments approach, random effects regression models without predictors were used to estimate prevalence k at the survey-arm level and incidence k at the study year-arm level. For the prevalence k, the mean prevalence and between-cluster variance were estimated for each survey arm $s$ using the following model:

$$x_{ijs} = \alpha_s + \upsilon_{js} + e_{ijs} \tag{7}$$

where $x_{ijs}$ is the observed malaria status (positive or negative) of $i$th individual in the $j$th cluster of survey arm $s$. The term $\alpha_s$ denotes the overall mean prevalence in survey arm $s$, while $\upsilon_{js}$ is the effect of the $j$th cluster on prevalence in survey arm $s$, and $e_{ijs}$ is the individual-level variation. The cluster effects $\upsilon_{js}$ follow a normal distribution with a mean 0 variance $\sigma_{B_s}^2$. Prevalence k and corresponding 95% confidence

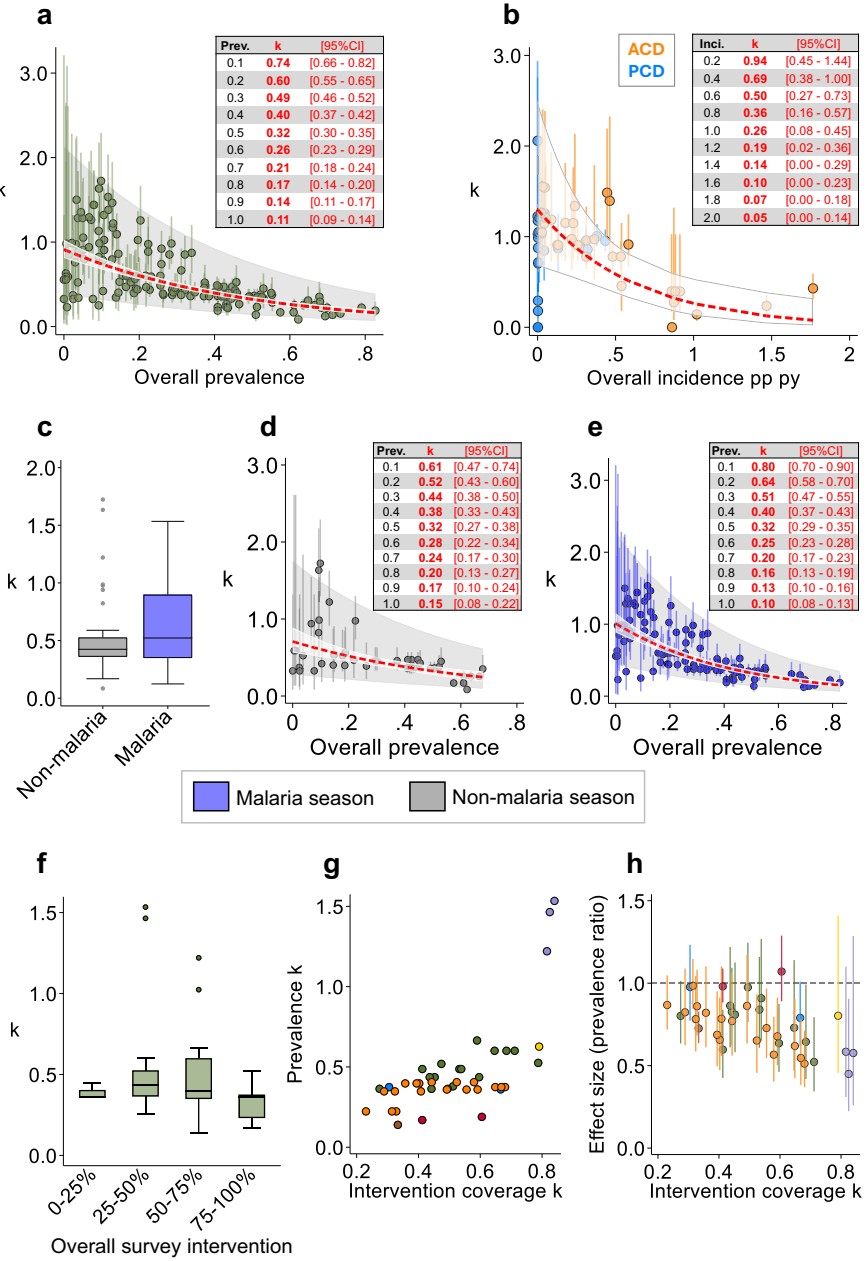

**Fig. 5 | Association between overall prevalence, incidence, seasonality and intervention coverage on between-cluster heterogeneity in malaria CRTs.**
**a** Relationship between overall survey-arm prevalence and observed (green dots) and predicted (red dash) prevalence k. Error bars: k 95%CIs. White shading: 95%CI of non-linear prediction. Grey shading: 95%PI of the survey observations.
**b** Relationship between overall study year-arm incidence and observed (orange dots) and predicted (red dash) incidence k based on ACD. Error bars: k 95%CIs. White shading: 95%CI of non-linear prediction. **c** Observed prevalence k estimates stratified by survey season: malaria, non-malaria. Box whiskers represent k estimate range. Box represents interquartile range of k estimates. Box horizontal line represents median k estimate. **d, e** Relationship between overall survey prevalence and observed prevalence k (dots) among trial surveys conducted in the non-

malaria (grey, D) and malaria (blue, E) season. Error bars: k 95%CIs. White shading: 95%CI of non-linear prediction. Grey shading: 95%PI of the survey observations. **f** Prevalence k estimates for each survey arm stratified by overall intervention coverage (%) among post-intervention survey arms. Box whiskers represent k estimate range. Box represents interquartile range of k estimates. Box horizontal line represents median k estimate. **g** Relationship between the cluster heterogeneity in prevalence and the cluster heterogeneity in intervention coverage among post-intervention surveys. Coloured dots represent surveys from the same trials. **h** Impact of between-cluster heterogeneity in intervention coverage on observed effect size estimates and uncertainty (95%CIs) among post-intervention surveys. Values beneath the dashed line show interventions that were shown to be effective. Coloured dots represent surveys from the same trials.

intervals for each survey arm were calculated from model outputs as follows:

$$Prevalence\ \hat{k}_s = \frac{\hat{\sigma}_{B_s}}{\hat{\alpha}_s} \qquad (8)$$

Corresponding 95% confidence intervals for *Prevalence* $\hat{k}_s$ were calculated based on the model-derived variance and its standard error:

$$95\%CI\ for\ Prevalence\ \hat{k}_s = \frac{\sqrt{\hat{\sigma}_{B_s}^2 \pm Z_{\alpha/2} \times SE(\hat{\sigma}_{B_s}^2)}}{\hat{\alpha}_s} \qquad (9)$$

## BOX 1

# Summary of recommendations and considerations for future malaria CRT design, conduct and analysis based on study findings

### CRT design considerations

**Choice of epidemiological outcome:** Cluster-level malaria incidence is more likely to exhibit larger between-cluster variability than malaria prevalence outcomes.

**Sample size estimation:**
- Incorporate a k/ICC estimate into your sample size calculation that is based on empirical baseline/prior data.
- In absence of baseline/prior cluster-level data, consider using an informed estimate of k based on the expected prevalence/incidence across the trial setting (suggested values are shown in Fig. 5a, b).
- In low-endemicity settings, be aware trials will likely be underpowered to detect small relative differences between study arms in the presence of large-between cluster variability.
- In low and medium-endemicity settings, be aware the degree of between-cluster changes over time, even in the control arm.

### CRT implementation considerations

**Strive for even intervention coverage across clusters**. Uneven intervention coverage across clusters is associated with larger between-cluster variability in outcomes.

### CRT analysis considerations

**Analysis strategies:**
- Consider analysis approaches that allow for differential between-cluster variability between study arms.
- In low endemicity trials with very skewed cluster-distributions, non-parametric analysis methods may be more appropriate than parametric methods.

**Between-cluster heterogeneity reporting**. Report empirical estimates of between-cluster variability across all trial arms and at different trial stages, in accordance with CONSORT guidelines, to help inform future sample size calculations.

where $Z_{\alpha/2}$ is the critical value from the standard normal distribution.

Using the same model output components, we estimated the survey-arm prevalence ICC, which quantifies the proportion of total variance (i.e. between-cluster and within cluster variation) attributable to between-cluster variation:

$$Prevalence\ \widehat{ICC}_s = \frac{\hat{\sigma}_{B_s}^2}{\hat{\sigma}_{B_s}^2 + \hat{\sigma}_{E_s}^2} \tag{10}$$

where $\hat{\sigma}_{E_s}^2$ represents within-cluster (residual) variance derived from the individual-level error term $e_{ijs}$. Corresponding 95%CIs were obtained using the "estat icc" command in STATA (v. 18) according to:

$$95\%CI\ for\ \widehat{ICC}_s = \widehat{ICC}_s \pm Z_{\alpha/2} \times SE(\widehat{ICC}_s) \tag{11}$$

where $SE(\widehat{ICC}_s)$ is the standard error of the ICC estimated via the delta method.

For the estimation of incidence k at the study year level $s$, we used a Poisson regression model with cluster-level random effects and no predictors to estimate the overall study-year arm incidence and variance between clusters according to:

$$\lambda_{ijs} = exp(\alpha_s) \times v_{js} \tag{12}$$

where $\lambda_{ijs}$ corresponds to the observed malaria status (positive or negative) of the $i$th individual in the $j$th cluster of study year arm $s$. Parameter $\hat{\alpha}_s$ represents the overall mean incidence across all clusters in study year $s$ and $v_{js}$ is the random effect of cluster $j$ on incidence. The $v_{js}$ effects assume a gamma distribution with a mean of 1 and a variance of $\hat{\alpha}_s'$. Based on this distribution, the standard deviation of lambda

across clusters in study year arm $s$ can be estimated according to:

$$SD(\lambda_{js}) = exp(\hat{\alpha}_s) \times SD(v_{js}) = \hat{\alpha}_s \times \sqrt{\hat{\alpha}_s'} \tag{13}$$

and the incidence coefficient of variation ($k$) in each study year arm $s$ is then estimated as:

$$Incidence\ \hat{k}_s = \frac{SD(\lambda_{js})}{\hat{\alpha}_s} = \frac{\hat{\alpha}_s \times \sqrt{\hat{\alpha}_s'}}{\hat{\alpha}_s} = \sqrt{\hat{\alpha}_s'} \tag{14}$$

The 95% confidence interval for incidence k was derived using the standard error of the variance parameter $\hat{\alpha}_s'$ as follows:

$$95\%CI\ for\ incidence\ \hat{k}_s = \sqrt{\hat{\alpha}_s' \pm Z_{\alpha/2} \times SE(\hat{\alpha}_s')} \tag{15}$$

STATA (v.18) do file code used estimate k is included in Supplementary data 1. STATA do file code used to estimate ICC is included in Supplementary data 2. Code is accompanied with simulated cluster-level prevalence data (Supplementary data 3) and incidence data (Supplementary data 4).

### Data analysis

Using unmatched methods described in[12,32], we calculated each trial's predicted study power (%) according to original predictions of k and control-arm prevalence/incidence using the STATA command "clustersampsi" (v.18). Using empirical estimates of k and control-arm incidence and prevalence, we recalculated observed study power for each trial year and survey, respectively. For both predicted and observed power calculations, all additional parameters remained identical: significance level, cluster size, cluster numbers and desired effect size (anticipated % relative reduction between arms).

We further explored the impact of between-cluster heterogeneity on study power at the 5% significance level for a hypothetical trial with 20 clusters per arm, a cluster size of 50 and an assumed control prevalence of either 10%, 50% or 90%. Using varying k estimates (range: 0.3-1.5) and effect sizes (1-prevalence ratio, range: 0-1) we calculated corresponding study power (%) and sample size (required clusters per arm).

Using trial data, we investigated the impact of observed between-cluster heterogeneity on observed effect sizes between study arms during the intervention periods of trials. For each post-intervention prevalence survey and incidence year, we compared observed k estimates with observed cluster mean prevalence and rate ratios, respectively, along with corresponding 95% confidence intervals. Effect sizes were estimated as follows:

$$Effect\ size = \frac{\bar{T}_1}{\bar{T}_0} \tag{16}$$

where $\bar{T}$ represents the mean, cluster-level, estimate of prevalence or incidence in the intervention (1) arm and control (0) arm. To estimate corresponding 95%CIs, we multiplied and divided effect size estimates by t-distributed error factors estimated according to:

$$Error\ factor = exp\left(t_{\nu, 0.025} \times \sqrt{V}\right) \tag{17}$$

where $V$ represents the variance of the prevalence or rate ratios:

$$V = \frac{S_1^2}{c_1 \bar{T}_1^2} + \frac{S_0^2}{c_0 \bar{T}_0^2} \tag{18}$$

where $S^2$ corresponds to the within study arm variance and $c$ signifies the number of clusters per arm.

In addition to arm-level effect sizes, we estimated cluster-level effect sizes for each post-intervention survey for prevalence and each trial year for incidence. Cluster-level prevalence ratios were estimated by dividing each intervention cluster prevalence values by the mean control-arm prevalence in the corresponding survey. Cluster-level incidence ratios were similarly estimated by dividing each intervention cluster incidence value by the mean incidence in the corresponding study year control arm.

Upon estimating k for each trial survey-arm, we investigated factors associated with elevated k. Random effects logistic regression models were used to generate odds ratios to estimate associations with elevated prevalence k surveys ($k > 0.5$). This threshold was chosen to dichotomise k as an estimate of 0.5 is considered conservative[32] and can result in large numbers of clusters in sample size estimations. Explanatory variables included overall survey-arm prevalence (<10%, 10–40%, >40%), mean cluster size (<60, >60), clusters per arm (<15, >15), study arm (control, intervention), season (malaria, non-malaria) and diagnostic (PCR, RDT, microscopy)). These explanatory variables were chosen as they represent key design considerations and were available for all included trials. All models were fit using maximum likelihood and included trial-level random effects as trials had multiple surveys. A multivariate model, constructed in a forward stepwise manner according to superior model fit (LRT < 0.05), was additionally used to generate adjusted odds ratios associated with elevated $k$ surveys to account for potential confounding by the above stated factors.

To further characterise the non-linear relationship between overall survey prevalence or study-year incidence and k, we conducted linear regression analyses on log-transformed values of k. In the prevalence models, the log-transformed k estimates at the survey-arm level served as the dependent variable, while overall survey-arm prevalence was the independent variable. For the incidence models, the dependent variable was the log-transformed study-year arm k estimate, with overall study-year incidence as the independent variable. Predicted k estimates were presented along with 95% confidence intervals (95% CIs) and 95% prediction intervals (95%PIs). The 95% CIs indicate the uncertainty around the linear prediction, whereas the 95%PIs capture the uncertainty around individual survey-arm or study-year arm observations. Data analyses were conducted in STATA version 18 (StataCorp, College Station, TX, USA).

### Reporting summary
Further information on research design is available in the Nature Portfolio Reporting Summary linked to this article.

## Data availability
Results from this meta-analysis include cluster-level data that were made freely available online at Clinical Epidemiology Resources (ClinEpiDB; https://clinepidb.org/ce/app/) or were provided directly. Provided datasets from previously published trials are owned by the authors listed in the supplementary information files. These data can be made available from the corresponding authors.

## Code availability
The STATA (v.17) statistical code used in this study to estimate the coefficient of variation (k) and intra-cluster correlation coefficient (ICC) for both prevalence and incidence outcomes are included in Supplementary data 1 and 2, respectively. Code is accompanied by fictious datasets including cluster-level prevalence data (supplementary data 3) and cluster-level incidence data (Supplementary data 4).

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

## Acknowledgements

This research is supported by a grant to the London School of Hygiene and Tropical Medicine and Imperial College London from the Bill & Melinda Gates Foundation (INV-038132). The funders had no role in study design, data collection and analysis, decision to publish or preparation of the manuscript. The authors wish to thank all the participants and study personnel of the trials included in this meta-analysis. Finally we would like to further thank all trial study teams for either providing data directly, or making datasets accessible online.

## Author contributions

J.B., J.D.C., T.S.C., and J.C. designed the study. E.E., C.C., F.S., S.G.S., S.V., J.B.C., M.S.H., E.D.D., A.E., U.DA., R.T., T.J.P., L.v.S., M.M., J.M., N.P., and M.A. were involved with the original trials and provided the cluster-level data directly and/or made the data freely available online. J.B. and J.C.D. conducted the analysis. D.D., T.S.C., J.C., and R.H. assisted with the analysis. T.S.C. and J.C. supervised the project. J.B. wrote the first draft of the manuscript. All authors discussed the results, edited the manuscript and approved the final version.

## Competing interests

All authors declare no competing interests.

### Ethics

This study is a meta-analysis of cluster-level data with no personal identifiers, derived exclusively from previously published studies. All trial data were obtained with the original study corresponding authors' consent or were made freely available online. No new data involving human participants were collected or analysed by the authors.

## Additional information

[1]International Statistics and Epidemiology Group, Department of Infectious Disease Epidemiology and International Health, London School of Hygiene and Tropical Medicine (LSHTM), London, UK. [2]Medical research Council (MRC) Centre for Global Infectious Disease Analysis, Department of Infectious Disease Epidemiology, Faculty of Medicine, Imperial College London, London, UK. [3]Centro de Investigaçao em Saúde de Manhiça, Manhiça, Mozambique. [4]Barcelona Institute for Global Health (ISGlobal), Barcelona, Spain. [5]CIBER de Enfermedades Infecciosas, Madrid, Spain. [6]Universidad de Navarra, Pamplona, Spain. [7]Department of Vector Biology,  Liverpool School of Tropical Medicine, Liverpool, UK. [8]Kenya Medical Research Institute (KEMRI) Centre for Global Health Research, Kisumu, Kenya. [9]National Malaria Control Programme, Ministry of Health, Manzini, Eswatini. [10]Department of Epidemiology and Population Health, Stanford University, Stanford, USA. [11]Chan Zuckerberg Biohub, San Francisco, USA. [12]Malaria Elimination Initiative, Institute for Global Health Sciences, University of California, San Francisco (UCSF), San Francisco, USA. [13]Department of Epidemiology and Biostatistics, University of California, San Francisco (UCSF), San Francisco, USA. [14]Medical Research Council (MRC) Unit The Gambia at the London School of Hygiene and Tropical Medicine (LSHTM), Disease Control & Elimination Theme, Fajara, The Gambia. [15]Mahidol Oxford Tropical Medicine Research Unit, Faculty of Tropical Medicine, Mahidol University, Bangkok, Thailand. [16]Centre for Tropical Medicine and Global Health, Nuffield Department of Medicine, University of Oxford, Oxford, UK. [17]Department of Parasitology, National Institute for Medical Research, Mwanza, Tanzania. [18]Department of Epidemiology and Public Health, Swiss Tropical & Public Health Institute, Basel, Switzerland. [19]University of Basel, Basel, Switzerland. [20]Population Services International (PSI), Malaria department, Cotonou, Benin. [21]Institut de Recherche Clinique du Benin (IRCB), Clinical Research department, Abomey-Calavi, Benin. ✉e-mail: joseph.biggs1@lshtm.ac.uk

