## [Peer Review file · Nature Communications]

Characterisation of between-cluster heterogeneity in malaria cluster randomised trials to inform future sample size calculations

Corresponding Author: Dr Joseph Biggs

Version 0:

Reviewer comments:

Reviewer #1

(Remarks to the Author)

This meta-analysis of 24 malaria CRTs highlights the impact of between-cluster heterogeneity on trial power and effect size precision, particularly in low endemicity settings. The authors conclude that many trials underestimated k , leading to insufficient power and reduced confidence in intervention effects.

I believe this work is a valuable contribution to the field and am not aware of any comparable work on a meta-analysis of malaria CRTs. The methodology seems sound and the authors established a framework across the 24 trials that allows for meaningful comparison.

I hence recommend accepting this manuscript for publication in Nature Communications with minor adjustments required.

I have several suggestions that could improve the quality:

- The formulation of the equations in the Methods section could be improved for clarity. I have highlighted specific areas for revision in the minor points.
- Given that the title suggests guidance for future sample size calculations, I recommend providing more explicit instructions for trialists on selecting an appropriate k .
- The manuscript would benefit from style refinements to address minor inconsistencies. I have outlined several in the minor points below.

Minor points:

101: "estimated without prior data" could be rephrased to assessed/chosen, since "estimated" implies the use of data.

111: Personally, I find the last sentence of the abstract to be too generic and disconnected from the results. How do the results of this work affect future trials?

129: The manuscript's readability would improve if citations stay within the same line as the relevant text.

131: CRTs assess the effectiveness of such tools, not the tools themselves.

190: In the systematic review, you report 71 malaria CRTs.

263: Dashed lines instead of Dash lines.

334: the a : is a different font than the superscript - makes it hard to understand

437: I would reduce "likely potentially" to one word

504: For completeness, I suggest providing more details on how you contacted the authors. Did you follow up if they did not respond? Additionally, do you have any insights into whether the 24 out of 71 trials included are representative?

514: I recommend removing "arbitrarily," as the selection appears to have followed a defined rule.

543: In the formulas 533/534 you define Prevalence s^2 and Incidence s^2 . In this formula you just use s^2 , making it unclear whether it refers to prevalence or incidence. For consistency and clarity, I suggest adapting the notation to distinguish between the two.

552: Similar to the comment above, there is no distinction between $\hat{\sigma}_B$ for prevalence and incidence.

556: If my understanding is correct, the "survey-arm mean and variance" were calculated separately for each arm, as indicated by the formula in line 558. Could you make this explicit by introducing an additional index for control and intervention? This would improve clarity. A similar notation is introduced below (line 600) using 1 and 0.

582: The title data analysis is quite generic

646: It seems that Richard Hayes is not listed in the author contributions

Supplementary: There is an inconsistency between capitalising "Supplementary Table" and "Supplementary table" in the titles.

Supplementary Table 7: (n) is defined twice. (Malaria positive, Intervention users)

(Remarks on code availability)

Reviewer #2

(Remarks to the Author)

Characterization of Between-Cluster Heterogeneity in Two Malaria Cluster Randomized Trials to Inform Future Sample Size Calculations

Comments

Abstract

1. "Results revealed that empirical estimates of k often exceeded those used in sample size calculations, significantly compromising study power and effect size precision. Increased between-cluster heterogeneity of outcomes was associated with specific outcome measures (i.e., incidence or prevalence), lower endemicity, survey seasonality, and uneven intervention coverage across clusters."

Comments: Provide supporting statistical evidence.

2. The conclusion in the abstract is not strong enough to reflect the topic, "to inform future sample size calculations." What specific guidelines does this study offer?

Methods

3. Did the authors screen and appraise all 24 trials? This is not explicitly addressed in the Methods section.

Results

4. "The review included 71 malaria CRTs that measured malaria-specific epidemiological outcomes (prevalence or incidence) and randomized at least six geographical clusters to study arms."

However, the authors also state: "Of the 74 malaria cluster-randomized trials (CRTs) identified in our systematic review, we obtained cluster-level epidemiological data from 24 trials after contacting corresponding authors (Supplementary Table 1)."

Comments: Which number is correct, 71 or 74?

5. Is 6539 in Table 2 is a valid statistic? Address this in the interpretation if it represents a real statistic, possibly due to sample size.

6. Define ACD and PCT in the footnote of Figure 1.

7. "Among all survey arms in all trials, prevalence k ranged from..." (Line 231 onward)

How do these results differ from the k values defined in Table 1? Are the studies listed in Table 1 a subset of the 24 studies? Why?

8. In Figure 2, how do the authors explain negative values for months since intervention?

9. The k differences presented in Figure 1D show that most results fall within ± 0.5 , which is not significantly problematic for sample size planning. On what basis do the authors conclude that most k values defined in sample size planning were overestimated?

Methods

10. "Prevalence was categorized as high (>40%), medium (10–40%), or low (0–10%). For incidence, endemicity was categorized by malaria cases per person-year (py) as high (>0.8/py), medium (0.2–0.8/py), or low (0–0.2/py)."

Comments: Since this review is based on RCTs, which do not aim to determine prevalence and incidence, researchers typically calculate the sample size beforehand. Additionally, this statement lacks clarity. When researchers report prevalence or incidence, they must first define the population characteristics.

What are the definitions of prevalence and incidence in this study? Did all 24 RCTs use the same definitions, referencing their respective populations? This issue is relevant to every mention of prevalence and incidence in the Results section. The authors should clarify population definitions before reporting prevalence and incidence values.

Data Analyses

10. "In the prevalence models, log-transformed k estimates at the survey-arm level served as the dependent variable, while overall survey-arm prevalence was the independent variable. For the incidence models, the dependent variable was the log-transformed study-year arm k estimate, with overall study-year incidence as the independent variable."

Why was k log-transformed when it ranges from 0.0 to 2.1? Was it severely skewed? Please provide justifications.

Other Concerns

11. This study makes a valuable contribution by validating k using completed RCTs. In sample size planning, researchers must set k a priori without knowing its true value once the trial concludes. This challenge is not unique to cluster trials but applies to all effect sizes and statistical tests. Given that k varies across studies, what concrete guidelines can the authors propose after reviewing and conducting the meta-analysis? Beyond stating that k is often underestimated/overestimated and presenting other findings, it would be highly beneficial to include a table outlining recommendations for setting k . These recommendations should be based on empirical findings without overstatements and thus offering fair, evidence-based guidance.

(Remarks on code availability)

Due to some technicality and IT facility problem, thus I am not able to validate the codes.

Reviewer #3

(Remarks to the Author)

This is an interesting study that looks at the between-cluster heterogeneity in the malaria cluster randomised trials. Although the aim of this manuscript is to investigate how the coefficient of variation (k) affects the power of a cluster randomised trial, I have somewhat different concerns:

1. This study consider ICC for incidence and prevalence, as it is closely related to the effect sample sizes in random effects model, which is the recommended method for analysing cluster trials. What about the heterogeneity in the treatment effects across different clusters? If we consider a cluster itself a small study, the between-cluster heterogeneity in treatment effects in a cluster randomised trial is equivalent to the heterogeneity in a meta-analysis. The heterogeneity will reduce the statistical power as well. It would be interesting to know which heterogeneity (in incidence/prevalence or in treatment effects) has a greater impact on statistical power of a cluster trial.

2. The authors observed that the k value is often larger than anticipated, and elevated k values compromised study power and reduced effect size precision. I am more concerned about the heterogeneity in the treatment effects between the clusters. Just like a meta-analysis of randomised trials, if the heterogeneity is large, the pooled estimate may be hard to interpret and misleading, as it does not represent the effect observed in most trials. Similarly, if the between-cluster heterogeneity in treatment effects is large, the average effects may be misleading or even meaningless because readers may believe it is transportable to any cluster in the trial.

Since the authors have obtained cluster-level data from many trials, I would be very interested in learning which heterogeneity is more important.

(Remarks on code availability)

As the data is not available, it is not possible to evaluate the code.

Version 1:

Reviewer comments:

Reviewer #2

(Remarks to the Author)

Congratulations to the authors for their hard work. What I liked most was Box 1—it really nailed the key points.

(Remarks on code availability)

Thank you

Reviewer #3

(Remarks to the Author)

Thank you for taking my comments into considerations when revising this manuscript. It is an interesting and important study. I have no further comments.

(Remarks on code availability)

Author's response to reviews

Dear Reviewers,

Re: "Characterisation of between-cluster heterogeneity in malaria cluster randomised trials to inform future sample size calculations"

We are resubmitting a revised version of our manuscript titled " Characterisation of between-cluster heterogeneity in malaria cluster randomised trials to inform future sample size calculations " for publication in Nature Communications. We thank reviewers for their useful insights and suggestions. We have responded to specific comments below and refer to changes in the revised manuscript. We hope this clarifies any issues raised and look forward to hearing from you.

Reviewer #1

Comment: This meta-analysis of 24 malaria CRTs highlights the impact of between-cluster heterogeneity on trial power and effect size precision, particularly in low endemicity settings. The authors conclude that many trials underestimated k , leading to insufficient power and reduced confidence in intervention effects. I believe this work is a valuable contribution to the field and am not aware of any comparable work on a meta-analysis of malaria CRTs. The methodology seems sound and the authors established a framework across the 24 trials that allows for meaningful comparison. I hence recommend accepting this manuscript for publication in Nature Communications with minor adjustments required. I have several suggestions that could improve the quality.

Comment: The formulation of the equations in the Methods section could be improved for clarity. I have highlighted specific areas for revision in the minor points.

Response: We thank reviewer #1 for their positive comments have made the changes to the formulae as suggested in the minor points below.

Comment: Given that the title suggests guidance for future sample size calculations, I recommend providing more explicit instructions for trialists on selecting an appropriate k .

Response: We agree with this comment and now provide specific recommendations for selecting k , (in addition to other malaria CRT design and analysis recommendations) in the discussion section. (See discussion section, Box 1). Below shows box 1:

"Box 1: Summary of recommendations and considerations for future malaria CRT design, conduct and analysis based on study findings..."

CRT design considerations

Choice of epidemiological outcome: Cluster-level malaria incidence is more likely to exhibit larger between-cluster variability than malaria prevalence outcomes.

Sample size estimation:

- Incorporate a k /ICC estimate into your sample size calculation that is based on empirical baseline/prior data.
- In absence of baseline/prior cluster-level data, consider using an informed estimate of k based on the expected prevalence/incidence across the trial setting (suggested values are shown in Fig. 5A&B)
- In low-endemicity settings, be aware trials will likely be underpowered to detect small relative differences between study arms in the presence of large-between cluster variability.
- In low and medium-endemicity settings, be aware the degree of between-cluster changes over time, even in the control arm.

CRT implementation considerations

Strive for even intervention coverage across clusters. Uneven intervention coverage across clusters is associated with larger between-cluster variability in outcomes.

CRT analysis considerations

Analysis strategies:

- Consider analysis approaches that allow for differential between-cluster variability between study arms.
- In low endemicity trials with very skewed cluster-distributions, non-parametric analysis methods may be more appropriate than parametric methods.

Between-cluster heterogeneity reporting. Report empirical estimates of between-cluster variability across all trial arms and at different trial stages, in accordance with CONSORT guidelines, to help inform future sample size calculations.

Comment: The manuscript would benefit from style refinements to address minor inconsistencies. I have outlined several in the minor points below.

Minor points:

Comment: 101: "estimated without prior data" could be rephrased to assessed/chosen, since "estimated" implies the use of data.

Response: We agree with this suggestion. The abstract has been changed and now reads: *"The coefficient of variation (k), a measure of such heterogeneity, is typically used in malaria CRTs yet is often predicted without prior data."* (See abstract).

Comment: 111: Personally, I find the last sentence of the abstract to be too generic and disconnected from the results. How do the results of this work affect future trials?

Response: We agree with this point raised and have made the last sentence more specific: *"Study findings can enhance the robustness of future malaria CRT sample size calculations"*

by providing informed k estimates based on expected prevalence or incidence, in the absence of cluster-level data.”

Comment: 129: The manuscript's readability would improve if citations stay within the same line as the relevant text.

Response: We recognise that placing citations on the same line as the relevant text does improve readability for this draft, however to make such changes would involve adding extra spaces between text and citations which may interfere with the final editing. This will however be something I address when I come to proofing the final draft.

Comment: 131: CRTs assess the effectiveness of such tools, not the tools themselves.

Response: We agree with this suggestion and have made changes to the introduction as follows: *“CRTs assess the effectiveness of such tools by randomising groups (clusters) into intervention and control arms, and estimating effect size(s) by comparing them [8].”* (See introduction)

Comment: 190: In the systematic review, you report 71 malaria CRTs.

Response: On line 190 it mistakenly read 74 trials. This mistake has been rectified. It now correctly reads 71 malaria CRTs.

Comment: 263: Dashed lines instead of Dash lines.

Response: This suggestion has been changed in all figure legend throughout the paper.

Comment: 334: the a: is a different font than the superscript - makes it hard to understand

Response: This table is now all in the same font as such it is easier to understand the subscript (See Supplementary Table 6).

Comment: 437: I would reduce "likely potentially" to one word

Response: This sentence no longer exists in the discussion in response to the changes suggested by other reviewers.

Comment: 504: For completeness, I suggest providing more details on how you contacted the authors. Did you follow up if they did not respond? Additionally, do you have any insights into whether the 24 out of 71 trials included are representative?

Response: As suggested we now include more details in the article on how we contacted authors for data requests. This sentence now reads as *“We sought cluster-level malaria outcome data from corresponding authors of CRTs identified in our previous systematic review (PROSPERO: CRD42022315741) [17]. Authors were initially approached by email, with a second follow-up email to non-responders”*

In addition, as suggested, we now investigate how representative the studies included in this meta-analysis are to those originally identified in the systematic review by comparing the trial characteristics which are displayed in a new table in the supplementary material. Overall

trials in this meta-analysis and those in the systematic review are broadly similar (see Supplementary Table 3). In the results section we now comment on this and state: “*The characteristics of trials in this meta-analysis closely resembled those from the previous systematic review, suggesting they form a representative sample (Supplementary Table 3).*”

Comment:514: I recommend removing "arbitrarily," as the selection appears to have followed a defined rule.

Response: We agree with this point raised and have changed the sentence as follows: “*We classified trial endemicity based on control-arm prevalence or incidence averaged during the entire trial. Trial endemicity according to prevalence was categorized as high (>40%), medium (10-40%), or low (0-10%). For incidence, trial endemicity was categorized by malaria cases per person-year (py) as high (>0.8/py), medium (0.2-0.8/py), or low (0-0.2/py).*” (see methods).

Comment: 543: In the formulas 533/534 you define Prevalence s^2 and Incidence s^2 . In this formula you just use s^2 , making it unclear whether it refers to prevalence or incidence. For consistency and clarity, I suggest adapting the notation to distinguish between the two.

Response: We agree with this comment regarding improving the clarity of the equations by specifying which s^2 (prevalence or incidence) is used in what equation. Equations 3 and 4 now read as follows:

To estimate the true between-cluster variance $\hat{\sigma}_B^2$ for each survey arm for prevalence (Equation 3) or study year-arm for incidence (Equation 4), we subtracted the random sampling error from the empirical variance s^2 as follows:

$$\text{Prevalence } \hat{\sigma}_B^2 = s_{prev}^2 - \frac{p(1-p)}{\bar{n}_H} \quad (3)$$

$$\text{Incidence } \hat{\sigma}_B^2 = s_{inci}^2 - \frac{r}{\bar{f}_H} \quad (4)$$

where p refers to the overall survey-arm malaria prevalence, \bar{n}_H is the harmonic mean of the total number of individuals n_i tested per cluster ($c/\sum(\frac{1}{n_i})$), r refers to the overall study year-arm malaria incidence and \bar{f}_H is the harmonic mean of the total follow up time in years y_i per cluster ($c/\sum(\frac{1}{y_i})$).

(see methods section).

Comment: 552: Similar to the comment above, there is no distinction between $\hat{\sigma}_B$ for prevalence and incidence.

Response: Likewise, this has also been changed for clarity. We now clearly distinguish $\hat{\sigma}_B$ for incidence and prevalence. Equations 5 and 6 now read as follows:

We then estimated prevalence k for each survey-arm (Equation 5) and incidence k for each study year arm (Equation 6) according to:

$$\text{Prevalence } \hat{k} = \frac{\hat{\sigma}_{B \text{ prev}}}{p} \quad (5)$$

$$\text{Incidence } \hat{k} = \frac{\hat{\sigma}_{B \text{ inci}}}{r} \quad (6)$$

Comment: 556: If my understanding is correct, the "survey-arm mean and variance" were calculated separately for each arm, as indicated by the formula in line 558. Could you make this explicit by introducing an additional index for control and intervention? This would improve clarity. A similar notation is introduced below (line 600) using 1 and 0.

Response: Reviewer #1 is correct, we derived the mean and variance from the model fitted at the survey-arm level. We agree that including an additional index in our equations improves clarity. We now include an 's' index in our equations that represent survey-arm (and study year s for incidence). See equation 7. For consistency, we also fitted this to the remaining (equations 8-15). Equations 7-9 now reads as follows:

In addition to the methods-of-moments approach, random effects regression models without predictors were used to estimate prevalence k at the survey-arm level and incidence k at the study year-arm level. For the prevalence k , the mean prevalence and between-cluster variance were estimated for each survey arm s using the following model:

$$x_{ijs} = \alpha_s + v_{js} + e_{ijs} \quad (7)$$

where x_{ijs} is the observed malaria status (positive or negative) of i th individual in the j th cluster of survey arm s . The term α_s denotes the overall mean prevalence in survey arm s , while v_{js} is the effect of the j th cluster on prevalence in survey arm s , and e_{ijs} is the individual-level variation. The cluster effects v_{js} follow a normal distribution with a mean 0 variance $\sigma_{B_s}^2$. Prevalence k and corresponding 95% confidence intervals for each survey arm were calculated from model outputs as follows:

$$\text{Prevalence } \hat{k}_s = \frac{\hat{\sigma}_{B_s}}{\hat{\alpha}_s} \quad (8)$$

Corresponding 95% confidence intervals for *Prevalence* \hat{k}_s were calculated based on the model-derived variance and its standard error:

$$95\%CI \text{ for Prevalence } \hat{k}_s = \frac{\sqrt{\hat{\sigma}_{B_s}^2 \pm Z_{\alpha/2} \times SE(\hat{\sigma}_{B_s}^2)}}{\hat{\alpha}_s} \quad (9)$$

where $Z_{\alpha/2}$ is the critical value from the standard normal distribution.

Comment: 582: The title data analysis is quite generic

Response: We agree this subsection title is generic. However no term can be used to encompass both power and risk factor analysis and the journal restricts the number of the subheadings allowed. In this section we make sure we clarify what analysis we refer to in each paragraph.

Comment: 646: It seems that Richard Hayes is not listed in the author contributions

Response: This mistake has been rectified and it now states: "R.H assisted with the analysis"

Comment: Supplementary: There is an inconsistency between capitalising "Supplementary Table" and "Supplementary table" in the titles.

Response: This error has been fixed through the article and supplementary information. They now read as 'Supplementary Table X'.

Comment: Supplementary Table 7: (n) is defined twice. (Malaria positive, Intervention users)

Response: The small n is defined twice in this table as it refers simply to the numbers. Individuals surveys (N) includes all those surveyed, Malaria positive (n) represents those surveyed who tested positive for malaria while Intervention users (n), this refers to those who were surveyed, who used the intervention (ie reported sleeping under a trial net for example).

Reviewer #2

Characterization of Between-Cluster Heterogeneity in Two Malaria Cluster Randomized Trials to Inform Future Sample Size Calculations

Comments

Abstract

1. "Results revealed that empirical estimates of k often exceeded those used in sample size calculations, significantly compromising study power and effect size precision. Increased between-cluster heterogeneity of outcomes was associated with specific outcome measures (i.e., incidence or prevalence), lower endemicity, survey seasonality, and uneven intervention coverage across clusters."

Comment: Provide supporting statistical evidence.

Response: We appreciate this comment from reviewer #2 that the abstract lacks supporting statistical evidence. However, according to Nature Comms guidelines, the abstract should be a 150 word (approx..) non-technical summary of findings. As such we didn't include specific statistical evidence as this would increase the word count and decrease the clarity. We did however change this section to make it more specific: "*Results revealed empirical estimates of k often exceeded those used in sample size calculations which reduced study power and effect size precision. Elevated k values were associated with incidence outcomes (compared to prevalence), lower endemicity settings, and uneven intervention coverage across clusters.*" (See abstract)

Comment: 2. The conclusion in the abstract is not strong enough to reflect the topic, "to inform future sample size calculations." What specific guidelines does this study offer?

Response: We agree with reviewer #2, the last sentence of the abstract needed to be more specific and provide clear guidelines for future malaria cluster randomised trials. The last sentence of the abstract now reads as: "*Study findings can enhance the robustness of future malaria CRT sample size calculations by providing informed k estimates based on expected prevalence or incidence, in the absence of cluster-level data.*" (see abstract)

In addition, in the discussion of this article, we now provide a summary of specific guidelines based on the findings of this study. (See discussion, box 1 in article & response to reviewer #1 on page 2).

Comment: Methods. 3. Did the authors screen and appraise all 24 trials? This is not explicitly addressed in the Methods section.

Response: Reviewer #2 raises an important question regarding whether included CRTs in this meta-analysis were appraised. The cluster-level data from 24 malaria CRTs included in

this study represent a subset of trials previously identified in our published systematic review. We now highlight how representative this subset is to the original trials identified in the systematic review (see supplementary table 3) and show they share similar characteristics.

For the purposes of this meta-analysis, we opted to not to appraise included articles as we adopted strict criteria for inclusion into the systematic review. To be included, studies had to represent a malaria CRT whereby at least 6 geographical clusters were truly randomised to the either the control or intervention(s) arms. Moreover, trials had to measure malaria-specific epidemiological outcomes (incidence /prevalence) based on accurate diagnosis (ie RDTs, PCR, microscopy) and not non-specific outcomes like anaemia (which could be attributed to conditions other than malaria). We now make this clearer in the methods section which now reads: *“We sought cluster-level malaria outcome data from corresponding authors of CRTs identified in our previous systematic review (PROSPERO: CRD42022315741) [18]. Authors were initially approached by email, with a second follow-up email to non-responders. The initial review included 71 malaria CRTs that qualified for inclusion if they measured malaria-specific, epidemiological outcomes (prevalence or incidence) and randomised at least six geographical clusters to study arms”*.

We did investigate whether tools such as the “Cochrane risk of bias Tool for CRTs” could be used to appraise our included CRTs. But we refrained for two main reasons. Firstly, some of appraisal criteria was the same as our inclusion criteria (e.g the randomisation process). Second, given our meta-analysis and previous systematic review covers recent and historical malaria CRTs, a lot of older trials included less information which prevents us for truly appraising them. For instance, ‘deviations from intended outcomes’ is a common appraisal criteria. This would be easier to assess in recent articles where there is more scope in the supplementary material to include such detail, however older trials typically lack this information. As such if we appraised studies, we would not accurately be able to do based on the information provided and older trials would likely get lower scores.

Comment: 4. Results. “The review included 71 malaria CRTs that measured malaria-specific epidemiological outcomes (prevalence or incidence) and randomized at least six geographical clusters to study arms.” However, the authors also state: “Of the 74 malaria cluster-randomized trials (CRTs) identified in our systematic review, we obtained cluster-level epidemiological data from 24 trials after contacting corresponding authors (Supplementary Table 1).”

Comments: Which number is correct, 71 or 74?

Response: Reviewer #2 was right to highlight this mistake. It now correctly reads 71

Comment: 5. Is 6539 in Table 2 is a valid statistic? Address this in the interpretation if it represents a real statistic, possibly due to sample size.

Response: Reviewer #2 was right to highlight this upper bound for the 95%CI is a mistake. It now reads correctly as 6.539. (See Supplementary Table 7).

Comment: 6. Define ACD and PCT in the footnote of Figure 1.

Response: As suggested, this has been incorporated into the legend of figure 1 which now reads: *“ACD: active case detection incidence. PCD: passive case detection incidence.”*

Comment: 7. “Among all survey arms in all trials, prevalence k ranged from...” (Line 231 onward)

How do these results differ from the k values defined in Table 1? Are the studies listed in Table 1 a subset of the 24 studies? Why?

Response: Yes, results in table 1 (now changed to Supplementary Table 6) describes analysis on a subset of 9 of the 21 trials. In this analysis, we are comparing k estimates among trials that measured both prevalence and incidence outcomes at the same time (most trials measured just one type of outcome). We show that among trials that measured both outcomes, k was mostly lower for prevalence outcomes compared to k for incidence outcomes. We now make this clearer in the results: “*Among malaria CRTs that measured both incidence and prevalence during overlapping time periods ($n=9$), the data show that control-arm prevalence k estimates were lower than incidence k values for 89% (8/9) of trials (Supplementary table 6).*”

Comment: 8. In Figure 2, how do the authors explain negative values for months since intervention?

Response: Months since intervention indicate the time since the intervention was rolled out in each of the trials. In many trials, baseline surveys were conducted prior to intervention roll out. So -2 months since intervention can be interpreted a survey conducted 2 months before the intervention was rolled out. The legend of figure 2 now reads “*Surveys conducted <0 months since intervention represent those conducted prior to intervention roll out.*”

Comment: 9. The k differences presented in Figure 1D show that most results fall within ± 0.5 , which is not significantly problematic for sample size planning. On what basis do the authors conclude that most k values defined in sample size planning were overestimated?

Response: Reviewer #2 queries whether a difference in k within ± 0.5 is that problematic for sample size calculations. It should be noted for most CRTs, typical k values included in trials range from 0.1 to roughly 0.8 and are not defined as percentages (in other fields, coefficient of variations are presented as percentages, yet that is not common practise in malaria CRTs). Therefore a k difference of 0.1 is analogous to a 10% difference based on a percentage scale. Another reason k values are not presented as percentages is because they can exceed 1 (as shown in Figure 1D). In our article we illustrate what slight changes in k does required larger sample sizes (see Figure 4). Slight changes in k from 0.3, 0.6, 0.9 to 1.2 all have huge impacts on the number of cluster required to achieve 80% power.

Secondly, Figure 1D refers to the empirical k values not the differences. Differences in k in this study refer to differences between arms (see Figure 2A&B). Regarding over/underestimation of k , in Figure 3A, we show the difference in k between what was predicted and what was observed. Here you can see there are substantial differences and that most predicted k estimates in sample size calculations were underestimated and we state “*Assuming trialists anticipated their predicted k estimates would remain constant throughout their trials, 72.5% (29/40) of prevalence k and 57.9% (11/19) of incidence k values were underestimated.*”

Comment: Methods. 10. “Prevalence was categorized as high (>40%), medium (10–40%), or low (0–10%). For incidence, endemicity was categorized by malaria cases per person-year (py) as high (>0.8/py), medium (0.2–0.8/py), or low (0–0.2/py).” Comments: Since this

review is based on RCTs, which do not aim to determine prevalence and incidence, researchers typically calculate the sample size beforehand. Additionally, this statement lacks clarity. When researchers report prevalence or incidence, they must first define the population characteristics. What are the definitions of prevalence and incidence in this study? Did all 24 RCTs use the same definitions, referencing their respective populations? This issue is relevant to every mention of prevalence and incidence in the Results section. The authors should clarify population definitions before reporting prevalence and incidence values.

Response: We appreciate this point raised by reviewer #2 concerning defining prevalence and incidence in this study. Firstly, we now recognise the above mentioned sentence is confusing. Here we are categorising trial endemicity based on their observed prevalence/incidence in the control arm for the entire trial. I.e. trials with an overall prevalence >40% in the control arm were considered to be high endemicity for prevalence. In the methods, this sentence has been altered and now reads: *“We classified trial endemicity based on control-arm prevalence or incidence averaged during the entire trial. Trial endemicity according to prevalence was categorized as high (>40%), medium (10-40%), or low (0-10%). For incidence, trial endemicity was categorized by malaria cases per person-year (py) as high (>0.8/py), medium (0.2-0.8/py), or low (0-0.2/py).”*

Second, regarding the definitions of prevalence and incidence in this study. Prevalence per cluster is the number individuals tested for malaria / total tested for malaria in each cluster. For incidence per cluster, this refers to the number of new cases of malaria / total person years at risk (active incidence) or total population at risk (passive incidence). This is made clearer in the methods: *“For trials measuring prevalence, we requested the number of malaria-positive individuals and total tested per cluster, study arm, and survey (Supplementary Table 8). For incidence measured via active case detection (ACD), we requested new malaria cases and total person-years at risk, and for passive case detection (PCD), new malaria cases and the population at risk, stratified by cluster, study arm, and trial year (Supplementary Table 9 & 10). These data were supplemented with covariates from published articles, including diagnostic method and age range tested. Malaria prevalence in this study referred to the number of individuals tested positive for malaria over the total number of individuals tested for malaria. Malaria incidence in this study referred to the number of new cases divided by the total person years at risk (ACD) or total population in each cluster at risk (PCD).”*

Reviewer #2 is correct, prevalence and incidence was measured differently for all trials. The diagnostics used to test participants varied (RDT, PCR microscopy), the age ranges of those tested varied (some trials tested those aged 0-10 years others tested all ages). As such measures of prevalence and incidence are not directly comparable between trials in this study and is reported as a limitation in the discussion, which reads: *“estimates of prevalence and incidence are not always comparable between trials, due to the use of different diagnostics and/or age ranges tested.”*

Lastly regarding incidence. It is true there are lots of definitions of incidence (risk/rate) and are measured over different scales (person months or person years). For this study, we requested the incidence data that is based on active case detection to be provided as new cases / person time in years. In some cases this would have involved converting person months to person years per cluster. This was done before the data was provided to us. For trials that measured passive case detection incidence (new cases / total population at risk) we distinguish this from ACD incidence throughout the article.

Comment: Data analyses. 10. “In the prevalence models, log-transformed k estimates at the survey-arm level served as the dependent variable, while overall survey-arm prevalence was the independent variable. For the incidence models, the dependent variable was the log-transformed study-year arm k estimate, with overall study-year incidence as the independent variable.” Why was k log-transformed when it ranges from 0.0 to 2.1? Was it severely skewed? Please provide justifications.

Response: As we observed a non-linear relationship between empirical estimates of k and the corresponding prevalence/incidence (see Figure 5A&B), we deemed it appropriate to first log-transform estimates of k to linearise this relationship. After log transformation, we observed a linear relationship between log- k and prevalence/incidence. Therefore, by exponentiating the predicted log- k from a linear regression model, we obtained predicted estimates of k for given prevalence/incidence values. We now make it clearer why k estimates were log transformed in the methods: “*To further characterise the non-linear relationship between overall survey prevalence or study-year incidence and k , we conducted linear regression analyses on log-transformed values of k . In the prevalence models, the log-transformed k estimates at the survey-arm level served as the dependent variable, while overall survey-arm prevalence was the independent variable. For the incidence models, the dependent variable was the log-transformed study-year arm k estimate, with overall study-year incidence as the independent variable.*”

Comment: Other Concerns. 11. This study makes a valuable contribution by validating k using completed RCTs. In sample size planning, researchers must set k a priori without knowing its true value once the trial concludes. This challenge is not unique to cluster trials but applies to all effect sizes and statistical tests. Given that k varies across studies, what concrete guidelines can the authors propose after reviewing and conducting the meta-analysis? Beyond stating that k is often underestimated/overestimated and presenting other findings, it would be highly beneficial to include a table outlining recommendations for setting k . These recommendations should be based on empirical findings without overstatements and thus offering fair, evidence-based guidance.

Response: We thank reviewer #2 for this comment and suggestion. We agree this article would benefit from providing specific guidelines for future malaria trials other than highlighting k estimates are typically underestimated in sample size calculations. We also agree this would best be achieved by providing a summary table. We now include a summary of study recommendations based on study findings. See discussion, box 1. (see similar response to reviewer #1 on page 2).

Comment: (Remarks on code availability): Due to some technicality and IT facility problem, thus I am not able to validate the codes.

Response: As reviewer #2 highlighted they were unable to evaluate the code used to estimate k . We now include a detailed word document containing the STATA code (with explanations) in the supplementary material. This is accompanied with fictitious cluster-level incidence and prevalence data for readers to attempt the code themselves. We also refer to this material in the article in the methods: “*STATA (v.18) do file code used estimate k is included in Supplementary data 1. STATA do file code used to estimate ICC is included in Supplementary data 2. Code is accompanied with simulated cluster-level prevalence data*”

(Supplementary data 3) and incidence data (Supplementary data 4).” (See Supplementary data 1-4)

Reviewer #3

Comment: This is an interesting study that looks at the between-cluster heterogeneity in the malaria cluster randomised trials. Although the aim of this manuscript is to investigate how the coefficient of variation (k) affects the power of a cluster randomised trial, I have somewhat different concerns:

1. This study consider ICC for incidence and prevalence, as it is closely related to the effect sample sizes in random effects model, which is the recommended method for analysing cluster trials. What about the heterogeneity in the treatment effects across different clusters? If we consider a cluster itself a small study, the between-cluster heterogeneity in treatment effects in a cluster randomised trial is equivalent to the heterogeneity in a meta-analysis. The heterogeneity will reduce the statistical power as well. It would be interesting to know which heterogeneity (in incidence/prevalence or in treatment effects) has a greater impact on statistical power of a cluster trial.

2. The authors observed that the k value is often larger than anticipated, and elevated k values compromised study power and reduced effect size precision. I am more concerned about the heterogeneity in the treatment effects between the clusters. Just like a meta-analysis of randomised trials, if the heterogeneity is large, the pooled estimate may be hard to interpret and misleading, as it does not represent the effect observed in most trials. Similarly, if the between-cluster heterogeneity in treatment effects is large, the average effects may be misleading or even meaningless because readers may believe it is transportable to any cluster in the trial.

Since the authors have obtained cluster-level data from many trials, I would be very interested in learning which heterogeneity is more important.

As the data is not available, it is not possible to evaluate the code.

Response: We thank reviewer #3 for their insightful comments and suggestions, they raise key concepts including the intra-cluster correlation and between-cluster heterogeneity of treatment effects which we now address in this manuscript. Taking each point in turn:

First, regarding the suggestion of incorporating ICC, we now include empirical estimates of ICC (in addition to empirical estimates of the coefficient of variation (k)) in this analysis and characterise it temporally for each trial and differentially between trial study arms. In Figure 1E, we characterise the prevalence ICC for each survey-arm for all included trials in this meta-analysis and state in the results: “*Prevalence ICC ranged between <0.01 and 0.40 with a median of 0.09*” (see section results). As the ICC represents a **relative** measure of between-cluster heterogeneity (Proportion of the total variance that can be attributed to between-cluster heterogeneity), we compared survey-arm prevalence ICC estimates to empirical k estimates (**absolute** measures of between-cluster variability representing the ratio of the standard deviation in cluster-level prevalences to the mean overall prevalence) (see Figure 1F). Interestingly, overall, no observable correlation was identified between these two related measures. Nonetheless, after stratifying by the overall survey-arm prevalence, we did identify a positive correlation. In survey-arms with an overall prevalence >10%, ICC and k were positively correlated. In contrast, in survey-arms with an overall prevalence <10%, larger disparities between ICC and k were observed. Even when the ICC was very low, k often exceeded 0.7. We now state this in our results: “*As k and ICC represent distinct measures of between-cluster variability, we compared them at the survey-*

arm level (Fig. 1F). Among survey-arms with a prevalence >10%, we observed a positive correlation between k and ICC. In contrast, among survey-arms with overall prevalences <10%, larger disparities were observed between k and ICC. When ICC estimates were near zero, k often exceeded 0.7." (see section results). We attribute this finding to the fact k is sensitive to slight changes in overall prevalence, particularly when the overall prevalence approaches zero. In the discussion we now state: " k estimates are sensitive to small changes in overall prevalence or incidence, particularly in low transmission settings where the denominator approaches zero [36]." (See section discussion).

Similarly to k , we explored how the prevalence ICC changes between repeated surveys in the control and intervention arms of malaria CRTs (See Figure 2E & Supplementary Figure 4D). We found that *"...prevalence ICC estimates among the control arms of repeated surveys were largest and more temporally variable in the trials conducted in medium endemicity settings (Fig. 2E). A trend similarly observed for intervention-arm ICC estimates (Supplementary Fig. 4D). Together, results illustrate that cluster-outcome heterogeneity changes over the course of malaria CRTs, regardless of intervention presence"* (see section results).

In summary we believe our paper has been improved by characterising between-cluster heterogeneity using both k and ICC. However, it should be noted the main focus of this article remains the coefficient of variation. This is because this measure of cluster heterogeneity is used in over 80% of the sample size calculations in malaria CRTs.

Regarding incidence ICC, as incidence in these included malaria trials refer to rates (new cases/person time at risk), we were unable to estimate the ICC for incidence as person-time is not an observable unit to estimate ICC [Thomson 2009]. We now state this in the methods: *"Methods-of-moments and mixed effects regression modelling approaches were used to estimate empirical values of prevalence k and ICC at the survey-arm level and incidence k at the study year-arm level according to methods described by Hayes and Moulton [12, 32]. We refrained from estimating incidence ICC values, as rates with person-time denominators lack a clearly defined unit of observation [36]."*

As reviewer #3 highlighted they were unable to evaluate the code used to estimate ICC/ k . We now include a detailed word document containing the STATA code (with explanations) in the supp material. This is accompanied with fictitious cluster-level incidence and prevalence data for readers to attempt. See Supplementary data 1-4. We also refer to this code/fictitious data in the methods section: *"STATA (v.18) do file code used estimate k is included in Supplementary data 1. STATA do file code used to estimate ICC is included in Supplementary data 2. Code is accompanied with simulated cluster-level prevalence data (Supplementary data 3) and incidence data (Supplementary data 4)."*

Lastly, reviewer #3 suggests that investigating between-cluster heterogeneity in treatment effects may be more relevant to investigate than the between-cluster heterogeneity in outcomes (Incidence/prevalence). Firstly, it is important to make clear that all the trials included in this meta-analysis are parallel CRTs, as stepped wedge and cross-over design are not common in malaria research. We now make this clearer at the beginning of the results section: *"Of the 71 malaria cluster-randomised trials (CRTs) identified in our previous systematic review, we obtained cluster-level epidemiological data from 24 trials (Supplementary Table 1). These parallel CRTs, conducted across 21 different countries between 2000 and 2021, evaluated..."*

In parallel CRTs, cluster are exclusively exposed to intervention or control conditions during the entire the post-intervention period. As such, disentangling between-cluster treatment

effect heterogeneity from between-cluster outcome heterogeneity cannot be achieved [Hemming 2018]. Hemming et al state that “correlations between observations within clusters might vary across treatment arms” and that certain interventions may induce cluster-heterogeneity or homogeneity. Therefore to investigate treatment effect heterogeneity in our article, we now characterise the differences in k and ICC between arms during the intervention stages of trials (See updated Figure 2A-C). Interestingly, we observed that k is typically larger in the intervention arms of malaria trials while the ICC is typically larger in the control arms of trials. We now refer to this in the updated results section: “*For prevalence, k values were typically higher in the intervention arm during the post-intervention period (Fig. 2A) while ICC estimates were often larger in the control arm (Fig. 2B). This pattern was impacted by the overall survey prevalence which showed the difference in k between arms was lower in high prevalence surveys while the ICC difference was lower in low prevalence surveys (Supplementary Fig. 3). For incidence outcomes, k was typically higher in control arms of trials (Fig 2C). Despite no observed clear arm-differential between-cluster heterogeneity patterns among trials, k and ICC estimates were rarely similar between arms.*”

We then discuss this contrasting result in the discussion section: “*Results also showed that between-cluster heterogeneity was rarely similar between study arms. We propose two main explanations for this observation. First, k estimates are sensitive to small changes in overall prevalence or incidence, particularly in low transmission settings where the denominator approaches zero [36]. Second, the malaria CRTs included in this review evaluated a range of intervention types, which may have induced either homogeneous or heterogeneous treatment effects across clusters [37]. Notably, for prevalence outcomes in intervention arm, the ICC was generally lower, while k was often higher compared to the control arm. This may reflect more consistent intervention use within clusters, promoting within-cluster homogeneity and lowering the ICC, while the reduction in mean prevalence in the intervention arm may have inflated k estimates [36]. Regardless, commonly used analytical methods in CRTs, including random effects regression modelling and generalised estimating equations, typically presume equal cluster variability across treatment arms [32]. Whether malaria CRTs should consider analysis methods that allow for arm-specific variances remains an area of continued investigation, however recent evidence suggests that standard methods remain robust despite differences in ICC values between arms [38].*”

Lastly, to further explore between-cluster treatment heterogeneity and its relationship with between cluster variability in outcomes, we characterised individual cluster-level effect sizes. Before, we just looked at what elevated between-cluster heterogeneity in outcomes does to arm-level effect sizes and showed that high k is associated large confidence interventions around arm-level effect sizes (Fig 3D). In addition to this, we now look at the impact of large between cluster variability on individual intervention cluster-level (CL) effect sizes. CL-effect sizes were estimated by dividing each intervention clusters outcomes by the mean outcome in the corresponding control arm (i.e. this generates a CL-prevalence/incidence ratio). We then characterised the distribution of these CL effect sizes by the overall between-between cluster heterogeneity in outcomes of the surveys that are situated within (Fig. 3E&F). Among surveys with lower k value (<0.3), the distribution of CL-effect sizes is normally distributed with a mean just below one. This tells us most clusters are typically seeing a reduction and that there is not much variation in treatment effects. In contrast, In surveys with large between-cluster variability in outcomes ($k>1.2$), CL-effects sizes are really skewed. This means clusters exhibited a range of differences compared to the average control.

We now refer to this is in the updated results: “*Similarly to arm-level effect sizes, we explored the impact of empirical k estimates on cluster-level effect sizes (intervention cluster outcome / overall outcome in the corresponding control arm). For prevalence outcomes,*

among surveys with lower k estimates (<0.3), intervention cluster-effect sizes were normally distributed below a prevalence ratio of 1. In contrast, among surveys with higher k estimates (>1.2), cluster-level effect sizes exhibited a zero-inflated right-skewed distribution, indicating intervention clusters exhibited either a very large or no difference compared to the mean control arm prevalence (Fig 3E). A similar pattern was observed for incidence outcomes (Fig 3F). These results demonstrate large between-cluster heterogeneity in outcomes is equivalent to large between-cluster variability in treatment effects.”

We thank reviewer #3 for raising the issue of between-cluster variability in outcomes and treatment effects. We have enjoyed looking into this topic and hope our changes answer your queries.

References:

Hemming K, Taljaard M, Forbes A. Modelling clustering and treatment effect heterogeneity in parallel and stepped-wedge cluster randomized trials. *Stat Med.* 2018 Mar 15;37(6):883-898. doi: 10.1002/sim.7553. Epub 2018 Jan 8. PMID: 29315688; PMCID: PMC5817269.

Thomson A, Hayes R, Cousens S. Measures of between-cluster variability in cluster randomized trials with binary outcomes. *Stat Med.* 2009 May 30;28(12):1739-51. doi: 10.1002/sim.3582. PMID: 19378266.